# MiniPLM: Knowledge Distillation for Pre-training Language Models

**Yuxian Gu**[1,2,*], **Hao Zhou**[2], **Fandong Meng**[2], **Jie Zhou**[2], **Minlie Huang**[1,†]
[1]The CoAI Group, Tsinghua University  [2]WeChat AI, Tencent Inc., China

## Abstract

Knowledge distillation (KD) is widely used to train small, high-performing student language models (LMs) using large teacher LMs. While effective in fine-tuning, KD during pre-training faces efficiency, flexibility, and effectiveness issues. Existing methods either incur high computational costs due to online teacher inference, require tokenization matching between teacher and student LMs, or risk losing the difficulty and diversity of the teacher-generated training data. In this work, we propose **MiniPLM**, a KD framework for pre-training LMs by refining the training data distribution with the teacher LM's knowledge. For efficiency, MiniPLM performs offline teacher inference, allowing KD for multiple student LMs without adding training costs. For flexibility, MiniPLM operates solely on the training corpus, enabling KD across model families. For effectiveness, MiniPLM leverages the differences between large and small LMs to enhance the training data difficulty and diversity, helping student LMs acquire versatile and sophisticated knowledge. Extensive experiments demonstrate that MiniPLM boosts the student LMs' performance on 9 common downstream tasks, improves language modeling capabilities, and reduces pre-training computation. The benefit of MiniPLM extends to larger training scales, evidenced by the scaling curve extrapolation. Further analysis reveals that MiniPLM supports KD across model families and enhances the pre-training data utilization. Our code, data, and models can be found at https://github.com/thu-coai/MiniPLM.

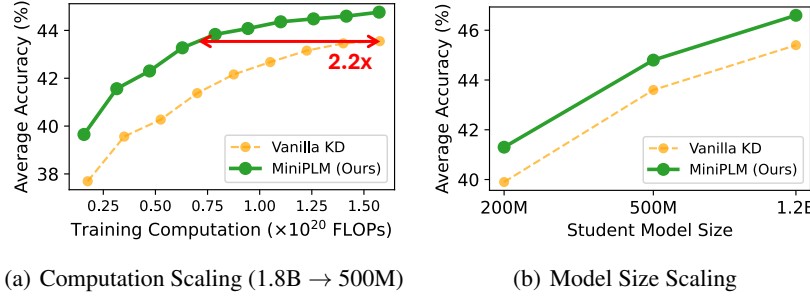

(a) Computation Scaling (1.8B → 500M)          (b) Model Size Scaling

Figure 1: Computation (a) and model size (b) scaling curves of student LMs pre-trained from scratch with Vanilla KD[1] and MiniPLM. The teacher LM has 1.8B parameters. "1.8B→500M" means we use a 500M student LM. Training-time computation is kept constant for LMs of the same size in model scaling. The y-axis represents the LMs' zero-shot performance on 9 downstream NLP tasks.

## 1 Introduction

Recent advances in language models (LMs; Han et al., 2021; Bommasani et al., 2021; OpenAI, 2023; Touvron et al., 2023a) have largely been driven by scaling up model sizes, but this comes with high inference costs for deployment. At the same time, training small, deployment-friendly LMs faces training computation challenges, as small models are typically far from compute-optimal

---

[*]Contribution during an internship at Tencent Inc. ⟨guyx21@mails.tsinghua.edu.cn⟩
[†]Corresponding author.

[1]Vanilla KD (Sanh et al., 2019; Muralidharan et al., 2024) minimizes the token-level forward Kullback-Leibler divergence between the output distributions of the teacher LM and student LM.

configurations according to Scaling Laws (Hoffmann et al., 2022). This has spurred a growing interest in exploring the limits of small LMs under the constraint of pre-training computation (Lu et al., 2024).

Knowledge Distillation (KD; Hinton et al., 2015), where a small student LM learns from a large teacher LM, is a promising approach for training high-performing small LMs. While KD is effective for fine-tuning (Xu et al., 2024), its role in improving pre-training, the critical stage for LMs to acquire foundation knowledge, remains under-explored. Applying KD during pre-training with the methods in fine-tuning stages is non-trivial, which can be categorized as **online KD** and **offline KD**. In online KD, the teacher LM has to perform inference during pre-training to provide token-level probability supervision[2] (Sanh et al., 2019; Gu et al., 2024b), introducing additional training-time overhead. As shown in Figure 2, while online KD enhances performance within the same training steps, its benefits diminish if the extra computation was instead used to extend pre-training without KD. In addition, most online KD methods require the teacher and student LMs to share

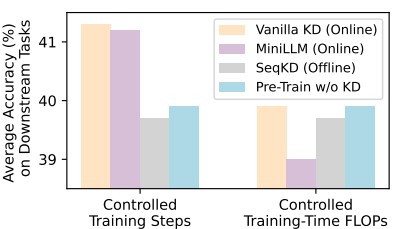

Figure 2: Results of applying KD methods in fine-tuning to pre-train a 200M student LM, using a 1.8B teacher LM. See Section 3.1 for method and evaluation details. When the training FLOPs are controlled, all KD methods perform similar or worse than Pre-Train w/o KD.

tokenization, limiting their flexibility for knowledge transfer across model families. Offline KD, on the other hand, avoids extra training-time computation and allows for KD across model families, as student LMs are trained on the data offline generated by the teacher (Kim & Rush, 2016; Gunasekar et al., 2023). However, ensuring sufficient difficulty and diversity in the generated data is challenging without extensive human expertise. Student LMs tend to overfit easy and common language patterns, hindering their downstream generalization (Shumailov et al., 2024), as shown in Figure 2.

To address these challenges, we propose **MINIPLM**, an efficient, flexible, and effective KD framework for pre-training LMs, as illustrated in Figure 3. MINIPLM typically works in scenarios where the teacher LM's probabilities are available to the users. Intuitively, it distills the teacher LM's knowledge into the pre-training distribution through *Difference Sampling*, which samples training instances based on the "difference" between large and small LMs. Student LMs are then pre-trained from scratch on the refined distribution. To ensure efficiency, as shown in Figure 3(a), *Difference Sampling* performs offline teacher LM inference, allowing MINIPLM to distill knowledge into multiple student LMs without incurring additional training-time costs. For flexibility, MINIPLM operates solely on the training corpus, enabling KD across model families and ensuring seamless integration with highly optimized training pipelines (Chowdhery et al., 2022). In terms of effectiveness, *Difference Sampling* samples training instances that the teacher LM prefers but that a small reference LM assigns low probabilities to, promoting data difficulty and diversity. As depicted in Figure 3(b), this design down-samples easy and common patterns, up-samples hard and diverse instances, and filters out noisy or harmful data points from the pre-training corpus, which encourages student LMs to acquire versatile and sophisticated knowledge, ultimately improving downstream generalization.

We apply MINIPLM to pre-train 200M, 500M, and 1.2B student LMs from scratch, using a 1.8B teacher LM. We show that MINIPLM surpasses various baselines in improving student LMs' zero-shot performance on 9 widely used downstream tasks, enhancing language modeling capabilities, and reducing pre-training computation. By extrapolating the test loss with the Scaling Law (Hoffmann et al., 2022), we observe that MINIPLM's benefit remains consistent for LMs trained on ~10T tokens. MINIPLM also facilitates KD across model families, improving Llama3.1 (Dubey et al., 2024) and Mamba (Gu & Dao, 2023) with a teacher LM from the Qwen family (Bai et al., 2023). Further analysis shows that MINIPLM enhances pre-training data utilization, reducing the data demand by 2.4 times, which mitigates the quick exhaustion of web-crawled corpora (Villalobos et al., 2022).

## 2 MINIPLM: KD FOR PRE-TRAINING LMS

We consider pre-training an LM with an output distribution $q_{\theta}$ on a large-scale corpus $\mathcal{D}$ consisting of $N$ text sequences, where $\theta$ represents the model parameters. KD aids pre-training by incorporating

---

[2]Pre-computing this supervision is impractical, as it requires 30PB of storage for 50B tokens with a 150K vocabulary. Therefore, teacher LM's probabilities are typically computed online during the student LM training.

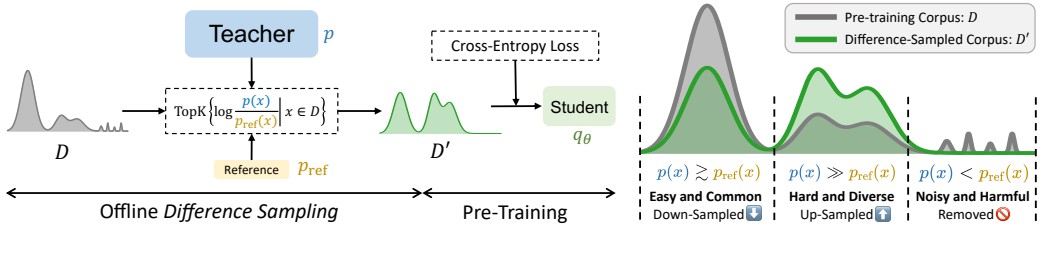

(a) MINIPLM Training Framework       (b) Effect of *Difference Sampling*

Figure 3: MINIPLM. (a): Training framework. MINIPLM distills the knowledge of the teacher LM into the student LM by adjusting the pre-training corpus of the student LM ($q_\theta$) through **offline** *Difference Sampling*, based on the output probability discrepancy between the teacher LM ($p$) and a small reference LM ($p_{\text{ref}}$). (b): Illustration of the effect of *Difference Sampling*, which down-samples common easy instances, up-samples hard valuable instances, and removes noisy harmful instances.

the knowledge of a teacher LM with output distribution $p$, and training $\theta$ to minimize the discrepancy between $p$ and $q_\theta$ (Sanh et al., 2019). MINIPLM formulates KD as a reward maximization problem (Haarnoja et al., 2017), which trains the student LM to generate diverse texts receiving high preference from the teacher LM (Section 2.1). To efficiently and effectively optimize the reward, as illustrated by Figure 3, MINIPLM employs *Difference Sampling* to refine the pre-training corpus (Section 2.2). This process improves the pre-training distribution in an offline manner with the teacher LM's knowledge and preserves the data diversity and difficulty with the help of a small reference model. The student LM is then pre-trained from scratch on the refined corpus (Section 2.3).

## 2.1 KD AS REWARD MAXIMIZATION

Recent works (Gu et al., 2024b; Agarwal et al., 2024; Ko et al., 2024) have shown the effectiveness of minimizing the reverse Kullback-Leibler divergence (KLD) between $p$ and $q_\theta$ for KD of LMs in the fine-tuning stage, which avoids $q_\theta$ from over-estimating the low-probability regions of $p$. We reformulate this objective as a reward maximizing problem (Haarnoja et al., 2017), which is suitable for designing efficient offline optimization methods (Levine et al., 2020):

$$\boldsymbol{\theta} = \arg\min_{\boldsymbol{\theta}} \text{KL}\left[q_{\boldsymbol{\theta}}||p\right] = \arg\min_{\boldsymbol{\theta}} \mathbb{E}_{\boldsymbol{x}\sim q_{\boldsymbol{\theta}}} \log \frac{q_{\boldsymbol{\theta}}(\boldsymbol{x})}{p(\boldsymbol{x})}$$
$$= \arg\max_{\boldsymbol{\theta}} \mathbb{E}_{\boldsymbol{x}\sim q_{\boldsymbol{\theta}}} r(p, q_{\boldsymbol{\theta}}, \boldsymbol{x}), \tag{1}$$

where the reward $r$ is defined as $r(p, q_{\boldsymbol{\theta}}, \boldsymbol{x}) = \log \frac{p(\boldsymbol{x})}{q_{\boldsymbol{\theta}}(\boldsymbol{x})}$. Intuitively, Eq. (1) trains $q_\theta$ to generate text $\boldsymbol{x}$ that receives high reward values, indicating a preference from the teacher LM (i.e., high $\log p(\boldsymbol{x})$ values), while also ensuring high diversity (i.e., low $\mathbb{E}_{\boldsymbol{x}\sim q_{\boldsymbol{\theta}}} q_{\boldsymbol{\theta}}(\boldsymbol{x})$ values). To optimize Eq. (1), a simple yet effective approach is Best-of-N (Stiennon et al., 2020; Bai et al., 2022; Touvron et al., 2023b), where a set $\mathcal{D}_{q_{\boldsymbol{\theta}}}$ containing $M$ candidates is first sampled from $q_\theta$ and $K$ instances with the highest rewards are selected from these candidates to form a new dataset $\mathcal{D}'_{q_{\boldsymbol{\theta}}}$:

$$\mathcal{D}'_{q_{\boldsymbol{\theta}}} = \text{top-}K\{r(p, q_{\boldsymbol{\theta}}, \boldsymbol{x}) | \boldsymbol{x} \in \mathcal{D}_{q_{\boldsymbol{\theta}}}\}, \tag{2}$$

where $\mathcal{D}_{q_{\boldsymbol{\theta}}} = \{x_m | x_m \sim q_{\boldsymbol{\theta}}, 1 \le m \le M\}$. The student LM is then trained on $\mathcal{D}'_{q_{\boldsymbol{\theta}}}$ to learn to generate texts with large $r(p, q_{\boldsymbol{\theta}}, \boldsymbol{x})$ values. Although Best-of-N achieves performing the teacher LM's inference prior to the student LM training similar to offline KD (Kim & Rush, 2016; Gunasekar et al., 2023), it still lacks the efficiency advantage of these methods because obtaining $\mathcal{D}'_{q_{\boldsymbol{\theta}}}$ requires (1) **sampling data from $q_\theta$** and (2) **computing the reward values with $q_\theta$**, making $\mathcal{D}'_{q_{\boldsymbol{\theta}}}$ non-transferable for pre-training other student LMs. It is also hard to ensure the diversity and difficulty of the $M$ candidates sampled from $q_\theta$ without careful prompt engineering with human expertise. In the following, we show that these issues can be effectively and efficiently addressed by *Difference Sampling*, which is the basis of the MINIPLM training algorithm.

## 2.2 DIFFERENCE SAMPLING

As shown in Figure 3(a), *Difference Sampling* refines the pre-training corpus $\mathcal{D}$ based on the discrepancy between $p$ and the output distribution $p_{\text{ref}}$ from a tiny reference LM, which eliminates the dependency of Eq. (2) on $q_\theta$, making the sampled corpus reusable in training multiple student LMs.

**Top-$K$ Sampling From $\mathcal{D}$, not $\mathcal{D}_{q_\theta}$.** To **avoid sampling data from $q_\theta$**, we sample instances with high $r(p, q_\theta, \boldsymbol{x})$ values from the pre-training corpus $\mathcal{D}$, rather than from the $\mathcal{D}_{q_\theta}$ generated by the student LM as in Eq. (2). This way, changing the student LM does not affect the candidate set, and $\mathcal{D}$ contains enough diverse and hard examples to be sampled for pre-training. The following proposition offers theoretical support for this approach, showing that the sampled training instances from $\mathcal{D}_{q_\theta}$ and $\mathcal{D}$ are highly likely to be the same when the sizes of $\mathcal{D}_{q_\theta}$ and $\mathcal{D}$ are sufficiently large:

**Proposition 2.1.** *Let $S$ be the sample space of two distributions $p_1$ and $p_2$, $\mathbf{X}_1, \mathbf{X}_2, \cdots, \mathbf{X}_N \sim p_1$ be $N$ i.i.d random variables, and $\mathbf{Y}_1, \mathbf{Y}_2, \cdots, \mathbf{Y}_M \sim p_2$ be $M$ i.i.d random variables. Let $r(\cdot) : S \mapsto \mathbb{R}$ be any injective function. Assume that $\forall \boldsymbol{x} \in S$, $p_1(\boldsymbol{x}) > 0$, $p_2(\boldsymbol{x}) > 0$. For a fixed $K$ satisfying $1 \le K \le \min\{N, M\}$, when $N \to +\infty$, $M \to +\infty$, we have*

$$P\left(\text{top-}K\left\{r(\mathbf{X}_n) \,|\, 1 \le n \le N\right\} = \text{top-}K\left\{r(\mathbf{Y}_m) \,|\, 1 \le m \le M\right\}\right) \to 1. \qquad (3)$$

The proof of Proposition 2.1 is provided in Appendix A. Here, $p_1$ represents the data distribution of $\mathcal{D}$, $p_2 = q_\theta$, which is the sampling distribution of $\mathcal{D}_{q_\theta}$, and $r(\boldsymbol{x}) = r(p, q_\theta, \boldsymbol{x})$. Intuitively, Proposition 2.1 reveals that when $|\mathcal{D}| = N$ and $|\mathcal{D}_{q_\theta}| = M$ are sufficiently large, the top-$K$ instances selected from both sets tend to overlap. This suits our scenario well, as $\mathcal{D}$ is typically large-scale, and it is advantageous to sample as many candidates from $\mathcal{D}_{q_\theta}$ as possible to find high-reward instances.

**Decoupling the Student and the Reward-Computing LM.** To **avoid computing reward values with $q_\theta$**, we replace it with $p_{\text{ref}}$, the output distribution of a tiny reference LM, typically smaller than the student LM, for reward computation. The reference LM is pre-trained on a small subset $\mathcal{D}_{\text{ref}}$, uniformly sampled from $\mathcal{D}$, allowing $p_{\text{ref}}$ to approximate $q_\theta$ with minimal computation. This is a reasonable approximation because $q_\theta$ evaluates the difficulty of instances, and the relative data difficulties generally remain consistent across different models (Ethayarajh et al., 2022). As a result, the reward function in Eq. (2) is replaced with $r(p, p_{\text{ref}}, \boldsymbol{x}) = \log \frac{p(\boldsymbol{x})}{p_{\text{ref}}(\boldsymbol{x})}$. In Appendix D.3, we empirically show that $r(p, p_{\text{ref}}, \boldsymbol{x})$ and $r(p, q_{\text{ref}}, \boldsymbol{x})$ have high correlations.

**In summary**, *Difference Sampling* constructs a pre-training corpus $\mathcal{D}'$ from $\mathcal{D} - \mathcal{D}_{\text{ref}}$ as follows:

$$\mathcal{D}' = \text{top-}K\left\{ \log \frac{p(\boldsymbol{x})}{p_{\text{ref}}(\boldsymbol{x})} \,\middle|\, \boldsymbol{x} \in \mathcal{D} - \mathcal{D}_{\text{ref}} \right\}, \qquad (4)$$

which is independent of $q_\theta$. As shown in Figure 3(b), *Difference Sampling* essentially refines the data distribution of $\mathcal{D}$ by comparing the "difference" between $p$ and $p_{\text{ref}}$, increasing difficulty and diversity while filtering out noise. A detailed discussion of these effects is provided in Section 2.5.

## 2.3 PRE-TRAINING ON DIFFERENCE-SAMPLED CORPUS

As illustrated in Figure 3(a), we pre-train the student LM from scratch on the difference-sampled corpus $\mathcal{D}'$ with the cross-entropy loss for next-token prediction, which is similar to standard pre-training. The loss function $L(q_\theta, \mathcal{D}')$ is given by

$$L(q_\theta, \mathcal{D}') = -\frac{1}{|\mathcal{D}'|} \sum_{\boldsymbol{x} \in \mathcal{D}'} \frac{1}{|\boldsymbol{x}|} \sum_{t=1}^{|\boldsymbol{x}|} \log q_\theta(x_t | \boldsymbol{x}_{<t}), \qquad (5)$$

where $|\boldsymbol{x}|$ is the length of $\boldsymbol{x}$, $x_t$ is the $t^{\text{th}}$ token, and $\boldsymbol{x}_{<t}$ denotes the prefix of $\boldsymbol{x}$ with $t-1$ tokens.

## 2.4 MINIPLM TRAINING PIPELINE

The general training pipeline of MINIPLM is as follows: (1) Uniformly sample a subset $\mathcal{D}_{\text{ref}}$ from the pre-training corpus $\mathcal{D}$, ensuring $|\mathcal{D}_{\text{ref}}| \ll |\mathcal{D}|$. (2) Train a reference LM from scratch on $\mathcal{D}_{\text{ref}}$ using the cross-entropy loss to obtain its output distribution $p_{\text{ref}}$. (3) Perform *Difference Sampling* with $p_{\text{ref}}$ and the teacher LM's output distribution $p$, generating a pre-training corpus $\mathcal{D}'$ from $\mathcal{D} - \mathcal{D}_{\text{ref}}$ using Eq. (4). The size of $\mathcal{D}'$ is controlled by a sampling ratio $\alpha$, where $K = \alpha|\mathcal{D} - \mathcal{D}_{\text{ref}}|$. (4) Pre-train student LMs from scratch on $\mathcal{D}'$ with the cross-entropy loss defined in Eq. (5).

## 2.5 DISCUSSION

**Efficiency and Flexibility of MINIPLM.** As shown in Figure 3(a), MINIPLM relies only on $p(\boldsymbol{x})$ and $p_{\text{ref}}(\boldsymbol{x})$, the teacher and reference LMs' probability over the entire sequence, which can be computed and stored offline because each instance in $\mathcal{D}$ is associated with a single floating-point number, amounting to only 200MB storage for 50B tokens with a 1,024 sequence length. Once $\mathcal{D}'$ is difference-sampled based on $p(\boldsymbol{x})$ and $p_{\text{ref}}(\boldsymbol{x})$, multiple student LMs can be efficiently pre-trained under the teacher LM's guidance without extra computational cost. In addition, this process modifies only the pre-training data, imposing no restrictions on the architecture or tokenization of the student LM, making MINIPLM highly flexible for integration into optimized pre-training frameworks (Chowdhery et al., 2022) and suitable for KD across model families. In contrast, online KD (Muralidharan et al., 2024; Gu et al., 2024b) relies on per-token distributions, which are infeasible to store offline, as it takes $50B \times 150K \times 4\text{byte} = 30\text{PB}$ storage for an LM with 150K vocabulary trained on 50B tokens, making online teacher LM inference necessary. Aligning per-token distributions also demands matching tokenizers between the teacher and student (Boizard et al., 2024), which complicates the re-implementation and optimization of pre-training workflows.

**Effectiveness of MINIPLM.** In essence, as illustrated in Figure 3(b), MINIPLM distills the teacher LM's knowledge into the student LM's pre-training distribution via *Difference Sampling*, producing three effects: (1) Down-sampling easy and common patterns that both the teacher and reference LM fit well, where $p(\boldsymbol{x}) \gtrsim p_{\text{ref}}(\boldsymbol{x})$ and $\log \frac{p(\boldsymbol{x})}{p_{\text{ref}}(\boldsymbol{x})} \gtrsim 0$. (2) Up-sampling hard and diverse knowledge, which the larger teacher LM has mastered but the smaller reference LM struggles with, and thus $p(\boldsymbol{x}) \gg p_{\text{ref}}(\boldsymbol{x})$ and $\log \frac{p(\boldsymbol{x})}{p_{\text{ref}}(\boldsymbol{x})} \gg 0$. (3) Discarding noisy and harmful instances that the teacher LM assigns lower probabilities to than the reference LM, where $\log \frac{p(\boldsymbol{x})}{p_{\text{ref}}(\boldsymbol{x})} < 0$. We provide examples of these effects in Appendix F. Training on the distribution with these effects encourages the student model to focus more on the sophisticated world knowledge learned by the teacher without being distracted by the noise. Note that without the comparison between $p$ and $p_{\text{ref}}$, the effect (1) disappears, leading to a pre-training corpus dominated by common patterns. This explains the effectiveness of MINIPLM against prior offline KD methods (Kim & Rush, 2016; Peng et al., 2023), where maintaining the data difficulty and diversity, critical to pre-training (Shumailov et al., 2024), is challenging without extensive human efforts on prompt engineering (Gunasekar et al., 2023).

## 3 EXPERIMENTS

### 3.1 EXPERIMENTAL SETUP

**Model.** We adopt the Qwen-1.5 (Bai et al., 2023) architecture in our experiments. We use the officially released 1.8B Qwen-1.5 model as the teacher LM and distill its knowledge into students with 200M, 500M, and 1.2B parameters. Detailed model configurations are provided in Appendix B.

**Pre-Training.** We construct pre-training corpora from the Pile (Gao et al., 2020). To control the computation in experiments, we pre-train all LMs on a maximum of 50B tokens, where documents are merged to construct instances with sequence lengths of 1,024. For online KD methods that incur additional train-time computation, we reduce their training steps to align the total training computation with pre-training without KD or offline KD methods. See Appendix B for more pre-training details.

**Baselines.** We compare MINIPLM with 4 baselines:
- **Pre-Train w/o KD** pre-trains the student LM on a 50B corpus uniformly sampled from the Pile dataset, without the guidance of the teacher LM's knowledge.
- **Vanilla KD** (Muralidharan et al., 2024) minimizes the token-level forward KLD between $p$ and $q_{\boldsymbol{\theta}}$, which requires **online** inference of the teacher LM to obtain the token-level output distributions.
- **SeqKD** (Kim & Rush, 2016) trains the student LM on the teacher-generated data. Since it is infeasible to generate all 50B tokens, we approximate Kim & Rush (2016) by using the first 768 tokens of each instance from the training corpus in **Pre-Train w/o KD** as the prompts and let the teacher LM generate the remaining tokens **offline**.
- **MiniLLM** (Gu et al., 2024b) minimizes the reverse KLD between $p$ and $q_{\boldsymbol{\theta}}$ with PPO (Schulman et al., 2017), which requires **online** inference of the teacher LM and **online** sampling

| | HS | LAM | Wino | OBQA | ARC-e | ARC-c | PIQA | SIQA | Story | Avg. |
|---|---|---|---|---|---|---|---|---|---|---|
| *1.8B Teacher → 200M Student* | | | | | | | | | | |
| Pre-Train w/o KD | 31.1 | 32.4 | 49.9 | **27.6** | 38.9 | 23.1 | 61.8 | 36.4 | 58.1 | 39.9 |
| Vanilla KD | 30.4 | 31.0 | **51.4** | 26.6 | 40.1 | 23.1 | 62.2 | 36.9 | 57.3 | 39.9 |
| MiniLLM | 30.2 | 29.4 | 50.0 | 26.6 | 39.0 | 21.3 | 60.5 | 36.6 | 57.6 | 39.0 |
| SeqKD | 30.5 | 31.0 | 51.3 | 27.4 | 39.3 | 22.4 | 61.3 | 36.9 | 57.4 | 39.7 |
| MINIPLM | **32.7** | **35.4** | **51.4** | 27.2 | **40.6** | **23.7** | **63.3** | **37.0** | **60.0** | **41.3** |
| *1.8B Teacher → 500M Student* | | | | | | | | | | |
| Pre-Train w/o KD | 35.8 | 40.1 | 51.0 | 30.2 | 41.7 | 24.4 | 65.4 | 38.2 | 61.4 | 43.2 |
| Vanilla KD | 37.0 | 39.9 | 51.7 | 29.4 | 45.1 | 24.2 | 65.8 | 38.0 | 61.6 | 43.6 |
| MiniLLM | 33.0 | 35.4 | 51.2 | 27.5 | 42.1 | 24.2 | 62.3 | 37.3 | 60.2 | 41.5 |
| SeqKD | 34.9 | 37.9 | 50.7 | 28.6 | 42.7 | 23.6 | 65.0 | 38.4 | 58.9 | 42.3 |
| MINIPLM | **39.0** | **42.6** | **52.2** | **30.2** | **45.8** | **24.9** | **67.0** | **39.0** | **62.2** | **44.8** |
| *1.8B Teacher → 1.2B Student* | | | | | | | | | | |
| Pre-Train w/o KD | 39.4 | 44.5 | 51.8 | 28.4 | 46.0 | 25.7 | 67.0 | 39.5 | 62.2 | 44.9 |
| Vanilla KD | 40.7 | 43.3 | 53.2 | 29.8 | 46.1 | 25.5 | 67.3 | 39.2 | 63.5 | 45.4 |
| MiniLLM | 36.1 | 42.5 | 51.2 | 28.5 | 44.1 | 25.3 | 65.8 | 37.9 | 61.4 | 43.6 |
| SeqKD | 38.5 | 41.4 | 51.9 | 29.2 | 46.5 | 25.1 | 66.3 | 39.0 | 61.0 | 44.3 |
| MINIPLM | **42.8** | **46.2** | **53.3** | **31.0** | **46.8** | **26.9** | **68.3** | **39.8** | **64.0** | **46.6** |

Table 1: Zero-shot accuracy scores on 9 widely-used downstream tasks and the average scores (Avg.). We use the Qwen-1.5 1.8B LM (Bai et al., 2023) as the teacher and Qwen LMs with 200M, 500M, and 1.2B parameters as the student. Student LMs with the same sizes consume the same training-time computation. The best scores of each model size are **boldfaced**.

from the student LM. We treat the first 768 tokens of the instances from the training corpus of **Pre-Train w/o KD** as the prompts and sample 256 tokens from $q_\theta$ during the exploration of PPO.

**MINIPLM.** We employ a 104M reference LM trained on 5B tokens. In Section 3.2 and 3.3, we consider a setting where $\mathcal{D}$ is sufficiently large, containing 105B tokens uniformly sampled from the Pile corpus. We reserve 5B tokens as $\mathcal{D}_{\text{ref}}$, and conduct *Difference Sampling* as per Eq. (4) on the other 100B tokens by setting $\alpha = 0.5$ to construct a 50B-token corpus $\mathcal{D}'$. In this way, the student LM is pre-trained on $\mathcal{D}'$ for one epoch. We use the loss difference of the teacher and reference LM to sampled instances from $\mathcal{D}$, which is equivalent to Eq. (4) as all instances have 1,024 tokens. In Section 3.4, we evaluate MINIPLM in a data-limited setting, where $\mathcal{D}$ is controlled to contain 50B tokens and the student LM is trained on the difference-sampled data for multiple epochs. See Appendix D.3 for the ablation studies on the reference LM and sampling ratio $\alpha$.

**Evaluation.** We assess the zero-shot accuracy of LMs trained with different methods on 9 downstream tasks widely used in examining the foundation abilities of base models (Touvron et al., 2023a; Groeneveld et al., 2024). We also test the language modeling capability of the LMs on a subset of DCLM (Li et al., 2024a), a high-quality corpus carefully curated with complex pipelines, to examine how well LMs capture broad and diverse knowledge. See Appendix B for more evaluation details.

## 3.2  MAIN RESULTS

**MINIPLM Improves Downstream Performance.** Table 1 shows zero-shot accuracy on downstream tasks for LMs trained by different methods, leading to three key observations. *First*, among all the baselines, only Vanilla KD outperforms Pre-Train w/o KD for relatively large student LM, given a constant computation budget. This highlights the room for improvement in KD for pre-training, especially when the gap between the teacher and the student is substantial. *Second*, MINIPLM-trained model achieves the best performance across most of the tasks. Compared to Pre-Train w/o KD, MINIPLM effectively leverages the teacher LM's knowledge to improve the student LM pre-training. Compared to online methods like Vanilla KD and MiniLLM, MINIPLM, as shown in Figure 3(a), incurs no additional training-time overhead, allowing the student LM to be optimized for more steps and leading to higher performance. Compared to offline methods like SeqKD, MINIPLM, as shown

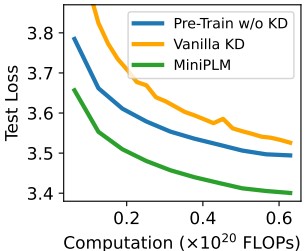 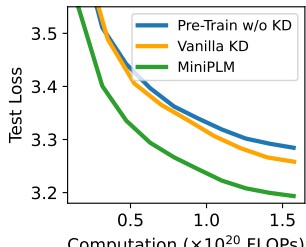 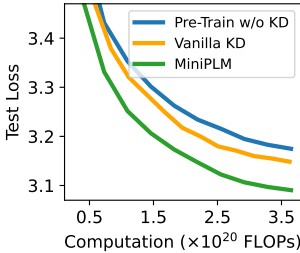

(a) 1.8B Teacher → 200M Student    (b) 1.8B Teacher → 500M Student    (c) 1.8B Teacher → 1.2B Student

Figure 4: Language modeling loss on the DCLM (Li et al., 2024a) subset. We distill the knowledge of the 1.8B Qwen model (Bai et al., 2023) into student LMs from the Qwen family with 200M, 500M, and 1.2B parameters. We control the total training-time FLOPs of different methods to be the same.

in Figure 3(b), uses a reference LM to ensure sufficient difficulty and diversity of the pre-training corpus, which is essential for the student LM to learn versatile sophisticated knowledge during pre-training and generalize across various downstream tasks. *Finally*, the improvements of MINIPLM against Vanilla KD scale well with the student LM size, which is also illustrated in Figure 1(b).

**MINIPLM Helps Language Modeling.** Figure 4, compares the language modeling performance of the MINIPLM-trained LMs and baselines on the DCLM (Li et al., 2024a) subset, a diverse and high-quality dataset curated from web corpora. The results show that, given the same training-time FLOPs, LMs trained with MINIPLM achieve the lowest test losses. In Table 2, we extrapolate the test losses with the Scaling Law (Hoffmann et al., 2022) to simulate pre-training on 1T and 10T tokens (details in Appendix D.2), showing thatMINIPLM maintains its advantages at the scales of pre-training recent large LMs (Touvron et al., 2023a; Dubey et al., 2024). A critical stage in DCLM's data-cleaning pipeline involves removing easy and common patterns, thereby enhancing challenging and diverse signals. Therefore, a lower test loss on DCLM suggests that, with the teacher LM's guidance and the reference LM, the MINIPLM-trained LMs learn the diverse and hard knowledge better due to the up-sampling of the corresponding parts in the pre-training distribution, as illustrated in Figure 3(b).

| $N_{stu}$ | Method | $L_{1T}$ | $L_{10T}$ |
|---|---|---|---|
| 200M | Pre-Train w/o KD | 3.35 | 3.32 |
| | Vanilla KD | 3.39 | 3.35 |
| | MINIPLM | **3.28** | **3.26** |
| 500M | Pre-Train w/o KD | 3.12 | 3.08 |
| | Vanilla KD | 3.12 | 3.07 |
| | MINIPLM | **3.06** | **3.04** |
| 1.2B | Pre-Train w/o KD | 2.98 | 2.94 |
| | Vanilla KD | 2.95 | 2.91 |
| | MINIPLM | **2.92** | **2.88** |

Table 2: Test loss predictions using the Scaling Law (Hoffmann et al., 2022). $N_{stu}$: the student LM size. $L_{1T}$, $L_{10T}$: the loss when Pre-Train w/o KD and MINIPLM process 1T and 10T tokens, with Vanilla KD consuming the same training FLOPs.

**MINIPLM Reduces Training Computation.** We plot the 500M student LM's average zero-shot accuracy scores on the downstream tasks in Figure 1(a) with respect to its pre-training FLOPs. MINIPLM achieves the same performance as Vanilla KD while reducing computational costs by 2.2 times. Similar trends are observed for other student LMs (Figure 7) and in the test loss curves on DCLM corpus (Figure 4). The efficiency gains are more pronounced when comparing MINIPLM with Pre-Train w/o KD on the 500M and 1.2B models. We attribute this acceleration to *Difference Sampling*, which down-samples common patterns and filters out noisy signals from the pre-training corpus, as shown in Figure 3(b). As a result, the model trained with MINIPLM avoids wasting computation on learning the easy knowledge quickly memorized during the early training stage and is less distracted by the noisy outliers that slow the convergence down.

### 3.3 KD ACROSS MODEL FAMILIES

A noticeable advantage of MINIPLM over Vanilla KD and MiniLLM is its flexibility to distill the knowledge of a teacher LM into student LMs with completely different tokenizers and architectures without additional strategies like Boizard et al. (2024). In Table 3, we illustrate the performance

|  | Llama3.1 | | Mamba | |
| --- | --- | --- | --- | --- |
|  | Acc. | Loss | Acc. | Loss |
| Pre-Train w/o KD | 41.0 | 3.52 | 41.6 | 3.24 |
| SeqKD | 40.8 | 3.54 | 41.0 | 3.27 |
| MINIPLM | **41.8** | **3.43** | **42.6** | **3.15** |

Table 3: Results of KD across model families. We use the teacher and reference LM from the Qwen family to distill the Llama3.1 and Mamba models. The average zero-shot accuracies on the downstream tasks and the losses on the DCLM corpus are reported. Note that Vanilla KD and MiniLLM cannot be applied when the teacher and student LMs use different tokenizations.

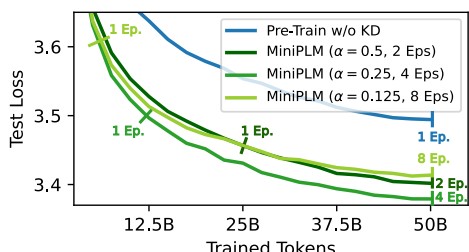

Figure 5: MINIPLM in the data-constrained setting. We fix $\mathcal{D}$ to contain 50B tokens and alter the sampling ratio $\alpha$ to obtain $\mathcal{D}'$ with *Difference Sampling*, which will be trained on for multiple epochs to achieve the constant total trained tokens. The y-axis represents the test loss on the DCLM corpus.

when using the LMs from the Qwen (Bai et al., 2023) family as the teacher and the reference LM to distill knowledge into a 212M Llama3.1 (Dubey et al., 2024) model and a 140M Mamba (Gu & Dao, 2023) model. The results demonstrate the promising performance of MINIPLM in KD across model families, outperforming Pre-Train w/o KD and the existing offline KD baseline (SeqKD). This allows emerging LMs with novel architectures (Lieber et al., 2024; Sun et al., 2023) or advanced tokenization (Tay et al., 2022; Godey et al., 2022) to inherit knowledge from existing LMs, thereby facilitating the development of more efficient and higher-performed models.

### 3.4 DATA-LIMITED SETTING

We evaluate MINIPLM in a data-limited setting where $\mathcal{D}$ is constrained to contain 50B tokens. To this end, the student LM should be trained on the difference-sampled $\mathcal{D}'$ over multiple epochs to ensure the total computation and trained tokens remain consistent with Pre-Train without KD. We split a $|\mathcal{D}_{\text{ref}}|$ containing 1B tokens to train the reference LM. Therefore, for a sampling ratio $\alpha$, the student LM should be trained for $\frac{|\mathcal{D}|}{\alpha|\mathcal{D}-\mathcal{D}_{\text{ref}}|} \approx \frac{1}{\alpha}$ epochs, given that $|\mathcal{D}_{\text{ref}}| \ll |\mathcal{D}|$. In Figure 5, we plot the loss curves of the 200M student LMs on the DCLM corpus when using $\alpha \in [0.5, 0.25, 0.125]$ and training the LM for around 2, 4, and 8 epochs, respectively. We can see that difference-sampling 25% data (with $\alpha = 0.25$) and training the student LM for 4 epochs yields the best performance, which aligns with the observations in Muennighoff et al. (2023). The corpus sampled with a higher $\alpha$ does not achieve the best quality and diversity offered by *Difference Sampling*, while a lower $\alpha$ leads to rapid over-fitting of the student LM. These findings suggest that MINIPLM is a promising approach to enhance data utilization when high-quality web corpora become scarce (Villalobos et al., 2022). By extrapolating the loss curve of Pre-Train w/o KD using the Scaling Law in a data-constrained setting (Muennighoff et al., 2023), we estimate that it would require an additional 68B training tokens to match the performance of MINIPLM ($\alpha = 0.25$, 4 epochs), which means MINIPLM reduces the pre-training data requirement by 2.4 times. See Appendix D.2 for more details on this extrapolation.

### 3.5 ANALYSIS

**Impact of Teacher Model.** Intuitively, larger teacher LMs will lead to better KD results, which is observed in recent works (Gu et al., 2024b). However, early works have also shown that an excessively large gap between teacher and student models can hinder effective KD (Mirzadeh et al., 2020). In Figure 6, we plot the performance of Vanilla KD and MINIPLM when distilling teacher LMs with different sizes into a 200M student model. We observe a similar phenomenon to Mirzadeh et al. (2020) on Vanilla KD and MINIPLM that larger teacher LMs are not necessarily more helpful for KD. However, we attribute this to different factors for Vanilla KD and MINIPLM. In Vanilla KD, the overhead introduced by larger LMs during training diminishes the benefits of distillation. As for MINIPLM, a 500M teacher LM proves most effective for distilling into a 200M student LM. Smaller teacher LMs (e.g., 300M) lack the capacity to identify the hard but valuable parts of the pre-training distribution, weakening the effectiveness of *Difference Sampling* as shown in Figure 3(b). On the

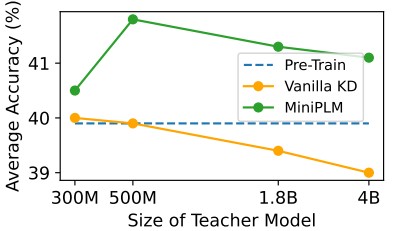

Figure 6: Impact of the teacher LM's sizes on Vanilla KD and MINIPLM, with the pre-training FLOPs aligned. The y-axis represents the average zero-shot accuracy on the downstream tasks.

| Pre-Training Corpus | Usage | Diversity |
|---|---|---|
| Original | Pre-Train w/o KD &Vanilla KD | 32.25 |
| Teacher-Generated | SeqKD | 30.16 |
| Difference-Sampled | MINIPLM | **36.70** |

Table 4: Semantic diversity of the original pre-training corpus $\mathcal{D}$ used in Pre-Train w/o KD and Vanilla KD, the teacher-generated corpus used in SeqKD, and the difference-sampled corpus $\mathcal{D}'$ in MINIPLM. *Difference Sampling* increases the diversity of the refined pre-training distribution, which helps LM pre-training.

other hand, the value scale of $\log p(\boldsymbol{x})$ from oversized LMs (e.g., 4B) becomes too small compared to that of the reference LM, which tends to degenerate *Difference Sampling* into sampling with $p_{\text{ref}}(\boldsymbol{x})$ only, losing the effect of the teacher LM. Future research could focus on optimizing teacher model size for MINIPLM or mitigating the impact of differing $\log p(\boldsymbol{x})$ and $\log p_{\text{ref}}(\boldsymbol{x})$ value scales.

**Diversity of Difference-Sampled Data.** To further verify the effectiveness of *Difference Sampling* on improving the diversity of the pre-training corpus, in Table 4, we follow Friedman & Dieng (2023) to compute the semantic diversity of the difference-sampled corpus and the data used in other baseline pre-training approaches. The results show that the difference-sampled corpus has the highest diversity, despite that *Difference Sampling* is derived from minimizing the reverse KLD, which exhibits the mode-seeking behavior (Minka et al., 2005). We suspect the reason is that *Difference Sampling* down-samples the easy parts of the corpus containing repeated contents while up-sampling the hard parts consisting of diverse texts. These two components, with large $p(\boldsymbol{x})$ values as seen in Figure 3(b), constitute the major modes of the teacher LM that $q_{\boldsymbol{\theta}}$ seeks during optimizing Eq. (1). Therefore, the mode-seeking behavior helps remove the noisy parts of the pre-training distribution, and the loss of diversity due to noise reduction is compensated by the up-sampling of the hard and diverse data points. We provide a case study in Appendix F to further explain our argument.

**Combining Vanilla KD and MINIPLM.** As shown in Table 1 and Figure 4, Vanilla KD improves the 500M and 1.2B student LM performance compared to Pre-Train w/o KD, suggesting the potential of further improving MINIPLM by combining it with Vanilla KD. This combined approach, termed "MINIPLM + Vanilla KD", applies Vanilla KD to the difference-sampled corpus used in MINIPLM. In Table 5, we compare "MINIPLM + Vanilla KD" with the individual use of Vanilla KD and MINIPLM, using a 1.8B teacher LM. The results show that when pre-training student LMs with 500B and 1.2B parameters, "MINIPLM + Vanilla KD" further improves the performance, given the same training-time FLOPs. This demonstrates that MINIPLM and Vanilla KD complement each other: MINIPLM distills the coarse-grained sequence-level knowledge of the teacher LM into the student LM via the pre-training data, while Vanilla KD directly aligns the token-level probability distribution between $p$ and $q_{\boldsymbol{\theta}}$, providing fine-grained token-level signals.

| $N_{\text{stu}}$ | Method | Acc. |
|---|---|---|
| 200M | Vanilla KD | 39.9 |
| | MINIPLM | **41.3** |
| | MINIPLM + Vanilla KD | 40.7 |
| 500M | Vanilla KD | 43.6 |
| | MINIPLM | 44.8 |
| | MINIPLM + Vanilla KD | **44.9** |
| 1.2B | Vanilla KD | 45.4 |
| | MINIPLM | 46.6 |
| | MINIPLM + Vanilla KD | **48.1** |

Table 5: Average accuracy on downstream tasks when combining MINIPLM and Vanilla KD. "MINIPLM + Vanilla KD": applying Vanilla KD to pre-train student LMs on the difference-sampled corpus in MINIPLM. $N_{\text{stu}}$: the size of student LMs.

## 4 RELATED WORK

**Language Model Pre-Training.** Pre-training is the critical phase for language models (LMs; Brown et al., 2020; OpenAI, 2023; Team et al., 2024; Chowdhery et al., 2022; Touvron et al., 2023a;b; Dubey

et al., 2024) to obtain their foundation abilities for various downstream tasks. To improve pre-training, some works focus on data curation, such as adjusting domain mixing (Xie et al., 2024; Ye et al., 2024), selecting valuable data points relevant to desired tasks (Brandfonbrener et al., 2024; Engstrom et al., 2024), or transforming the instances based on downstream requirements (Cheng et al., 2024; Gu et al., 2022; 2023). Another line of work improves the optimization during pre-training by solving better data reweighting strategies (Gu et al., 2024a), designing more effective optimizers (Liu et al., 2024; Shazeer & Stern, 2018), or discovering better training recipes (Hu et al., 2024; Hoffmann et al., 2022). Variations in model architectures (Xiong et al., 2020) and training objectives (Tay et al., 2023) are also explored to boost pre-training stability and final LM performance. In this work, we utilize the knowledge from existing LMs to enhance pre-training.

**Small Language Models.** Given the high computational demands of large LMs during inference, there has been growing interest in pre-training small LMs (Sardana et al., 2024). However, achieving high performance with limited parameter sizes remains challenging because the training computation that small LMs need to match the capabilities of large LMs often exceeds Chinchilla-optimal (Hoffmann et al., 2022) and scales as a power law with respect to the model size gap (Kaplan et al., 2020). Despite these challenges, recent efforts have made promising progress by data quality improvement (Mehta et al., 2024; Bellagente et al., 2024; Zhang et al., 2024) or model pruning (Muralidharan et al., 2024; Xia et al., 2024). We explore knowledge distillation as a complementary approach.

**Knowledge Distillation.** Knowledge distillation (KD; Hinton et al., 2015) uses a large teacher model to improve the performance of a small student model, which is widely used to build efficient neural network systems (Park et al., 2019; Czarnecki et al., 2019; Salimans & Ho, 2022). In NLP, early works primarily apply KD to encoder-only models (Devlin et al., 2019; Liu et al., 2019) for text classification by aligning token-level distribution (Sanh et al., 2019), hidden states (Sun et al., 2019), and attention matrices (Wang et al., 2020; 2021). For generative LMs, a straightforward KD approach is training small LMs on the texts generated by large LMs (Chiang et al., 2023; Peng et al., 2023; Hsieh et al., 2023). Other works (Gu et al., 2024b; Agarwal et al., 2024; Wen et al., 2023; Li et al., 2024b; Wu et al., 2024) explore better optimization objectives. However, these works focus on KD for fine-tuning LMs, while pre-training is critical for establishing core LM capabilities (Allen-Zhu & Li, 2024). Therefore, we investigate KD in the pre-training stage to develop strong small base LMs.

## 5 CONCLUSION

**Summary.** In this work, we find it non-trivial to adapt existing KD approaches for fine-tuning LMs to the pre-training stage because of the high overhead brought by the teacher LM inference in online KD and the tendency of losing data difficulty and diversity in offline KD. Therefore, we propose MINIPLM to address these issues through *Difference Sampling*, which refines the training distribution by down-sampling easy patterns, up-sampling hard instances, and filtering out noisy data points, with the knowledge of the difference between the large teacher LM and a small reference LM. The offline nature of MINIPLM makes it both efficient and flexible to distill student LMs with diverse configurations. The use of the large-small-model differences ensures the difficulty and diversity of the refined pre-training distribution. Using a 1.8B LM as the teacher to guide the pre-training of 200M, 500M, and 1.2B LMs, we demonstrate that MINIPLM improves the student LMs' performance on 9 downstream tasks, enhances their language modeling capability, and reduces pre-training computation. Additionally, MINIPLM improves the utilization of limited pre-training data and can distill teacher LM's knowledge into student LMs from completely different families.

**Limitations.** One limitation of MINIPLM is that it requires the large LMs' probabilities on texts from the pre-training corpus, making black-box KD for close-source LMs (OpenAI, 2023; Team et al., 2023) with MINIPLM challenging. For some APIs (OpenAI, 2022), this issue can be solved by specifying user-provided bias in the softmax operation, which allows obtaining the probabilities of given tokens one by one (Carlini et al., 2024), at the expense of a large number of API calls.

**Future Work.** A promising future direction to explore is applying the difference-sampled corpus to pre-train LMs larger than the teacher LM, enabling weak-to-strong generalization (Burns et al., 2023). Since data properties are critical for pre-training LMs across various sizes, the improvement of data diversity and difficulty is likely to be beneficial for LMs larger than the difference-sampling models.

ACKNOWLEDGEMENTS

This work is supported by the National Science Foundation for Distinguished Young Scholars (with No. 62125604) and Tsinghua University Initiative Scientific Research Program.

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

## A  PROOF OF PROPOSITION 2.1

To prove Proposition 2.1, We start from the $K = 1$ case, corresponding to selecting an instance with the maximal reward and Eq. (3) becomes

$$P\left(\arg\max_{1\leq n\leq N} r(\mathbf{X}_n) = \arg\max_{1\leq m\leq M} r(\mathbf{Y}_m)\right) \to 1, \tag{6}$$

which is equivalent to

$$\sum_{\boldsymbol{x}\in S} P\left(\arg\max_{1\leq n\leq N} r(\mathbf{X}_n) = \boldsymbol{x} \text{ and } \arg\max_{1\leq m\leq M} r(\mathbf{Y}_m) = \boldsymbol{x}\right) \to 1. \tag{7}$$

Since $\mathbf{X}_n$ and $\mathbf{Y}_m$ are independent random variables for $1 \leq n \leq N$ and $1 \leq m \leq M$, Eq. (7) can be further written as

$$\sum_{\boldsymbol{x}\in S} P\left(\arg\max_{1\leq n\leq N} r(\mathbf{X}_n) = \boldsymbol{x}\right) P\left(\arg\max_{1\leq m\leq M} r(\mathbf{Y}_m) = \boldsymbol{x}\right) \to 1. \tag{8}$$

We focus on the term $P\left(\arg\max_{1\leq n\leq N} r(\mathbf{X}_n) = \boldsymbol{x}\right)$, which can be expanded as follows based on the fact that $\mathbf{X}_1, \mathbf{X}_2, \cdots, \mathbf{X}_N$ are i.i.d random variables:

$$
\begin{aligned}
&P\left(\arg\max_{1\leq n\leq N} r(\mathbf{X}_n) = \boldsymbol{x}\right) \\
&= P\left(r(\mathbf{X}_n) \leq r(\boldsymbol{x}), \text{ for } 1 \leq n \leq N \text{ and at least one variable in } \mathbf{X}_1, \mathbf{X}_2, \cdots, \mathbf{X}_N \text{ equals } \boldsymbol{x}\right) \\
&= P\left(r(\mathbf{X}_n) \leq r(\boldsymbol{x}), \text{ for } 1 \leq n \leq N\right) - P\left(r(\mathbf{X}_n) \leq r(\boldsymbol{x}) \text{ and } \mathbf{X}_n \neq \boldsymbol{x}, \text{ for } 1 \leq n \leq N\right) \\
&= \prod_{n=1}^{N} P(r(\mathbf{X}_n) \leq r(\boldsymbol{x})) - \prod_{n=1}^{N} \left[P(r(\mathbf{X}_n) \leq r(\boldsymbol{x})) - P(\mathbf{X}_n = \boldsymbol{x})\right] \\
&= P(r(\mathbf{X}_1) \leq r(\boldsymbol{x}))^N - \left[P(r(\mathbf{X}_1) \leq r(\boldsymbol{x})) - p_1(\boldsymbol{x})\right]^N \\
&= P_{\mathbf{X}}(\boldsymbol{x})^N \left[1 - \left[1 - \frac{p_1(\boldsymbol{x})}{P_{\mathbf{X}}(\boldsymbol{x})}\right]^N\right],
\end{aligned} \tag{9}
$$

where $P_{\mathbf{X}}(\boldsymbol{x}) = P(r(\mathbf{X}_1) \leq r(\boldsymbol{x}))$. Note that $0 < p_1(\boldsymbol{x}) = P(\mathbf{X}_1 = \boldsymbol{x}) \leq P(r(\mathbf{X}_1) \leq r(\boldsymbol{x})) = P_{\mathbf{X}}(\boldsymbol{x})$, we have $0 < \frac{p_1(\boldsymbol{x})}{P_{\mathbf{X}}(\boldsymbol{x})} \leq 1$, and thus $\lim_{N\to+\infty} \left[1 - \frac{p_1(\boldsymbol{x})}{P_{\mathbf{X}}(\boldsymbol{x})}\right]^N = 0$. Similarly, by setting $P_{\mathbf{Y}}(\boldsymbol{x}) = P(r(\mathbf{Y}_1) \leq r(\boldsymbol{x}))$, we have

$$P\left(\arg\max_{1\leq m\leq M} r(\mathbf{Y}_m) = \boldsymbol{x}\right) = P_{\mathbf{Y}}(\boldsymbol{x})^M \left[1 - \left[1 - \frac{p_2(\boldsymbol{x})}{P_{\mathbf{Y}}(\boldsymbol{x})}\right]^M\right], \tag{10}$$

and $\lim_{M\to+\infty} \left[1 - \frac{p_2(\boldsymbol{x})}{P_{\mathbf{Y}}(\boldsymbol{x})}\right]^M = 0$. Let $\boldsymbol{x}^* = \arg\max_{\boldsymbol{x}\in S} r(\boldsymbol{x})$. Therefore, $P_{\mathbf{X}}(\boldsymbol{x}^*) = P_{\mathbf{Y}}(\boldsymbol{x}^*) = 1$ and $P_{\mathbf{X}}(\boldsymbol{x}) < 1, P_{\mathbf{Y}}(\boldsymbol{x}) < 1$ for $\boldsymbol{x} \neq \boldsymbol{x}^*$. When $N, M \to +\infty$, we have $P_{\mathbf{X}}(\boldsymbol{x}^*)^N = P_{\mathbf{Y}}(\boldsymbol{x}^*)^M = 1$, and $P_{\mathbf{X}}(\boldsymbol{x}^*)^N, P_{\mathbf{Y}}(\boldsymbol{x}^*)^M \to 0$ for $\boldsymbol{x} \neq \boldsymbol{x}^*$. Therefore, we have

$$
\begin{aligned}
&\lim_{N,M\to+\infty} \sum_{\boldsymbol{x}\in S} P\left(\arg\max_{1\leq n\leq N} r(\mathbf{X}_n) = \boldsymbol{x}\right) P\left(\arg\max_{1\leq m\leq M} r(\mathbf{Y}_m) = \boldsymbol{x}\right) \\
&= \sum_{\boldsymbol{x}\in S} \lim_{N,M\to+\infty} P_{\mathbf{X}}(\boldsymbol{x})^N P_{\mathbf{Y}}(\boldsymbol{x})^M \lim_{N,M\to+\infty} \left[1 - \left[1 - \frac{p_1(\boldsymbol{x})}{P_{\mathbf{X}}(\boldsymbol{x})}\right]^N\right] \left[1 - \left[1 - \frac{p_2(\boldsymbol{x})}{P_{\mathbf{Y}}(\boldsymbol{x})}\right]^M\right] \\
&= \lim_{N,M\to+\infty} P_{\mathbf{X}}(\boldsymbol{x}^*)^N P_{\mathbf{Y}}(\boldsymbol{x}^*)^M + \sum_{\boldsymbol{x}\in S, \boldsymbol{x}\neq\boldsymbol{x}^*} \lim_{N,M\to+\infty} P_{\mathbf{X}}(\boldsymbol{x})^N P_{\mathbf{Y}}(\boldsymbol{x})^M \\
&= 1,
\end{aligned} \tag{11}
$$

which proves Eq. (6). For $K > 1$, the equality of two top-$K$ subsets requires the elements ranked from 1 to $K$ to be equal, respectively. Therefore, the left hand of Eq. (3) can be decomposed using Bayes's Law. Let $\mathcal{X} = \{\mathbf{X}_1, \mathbf{X}_2, \cdots, \mathbf{X}_N\}$, $\mathcal{Y} = \{\mathbf{Y}_1, \mathbf{Y}_2, \cdots, \mathbf{Y}_M\}$, $\mathbf{X}^* = \arg\max\limits_{1 \le n \le N} r(\mathbf{X}_n)$, and $\mathbf{Y}^* = \arg\max\limits_{1 \le m \le M} r(\mathbf{Y}_m)$, we have $P(\mathbf{X}^* = \mathbf{Y}^*) \to 1$ (Eq. (6)) and

$$
\begin{aligned}
&P\left(\text{top-2}\left\{r(\mathbf{X}_n) \,|\, 1 \le n \le N\right\} = \text{top-2}\left\{r(\mathbf{Y}_m) \,|\, 1 \le m \le M\right\}\right) \\
&= P\left(\arg\max_{\mathbf{X} \in \mathcal{X} - \{\mathbf{X}^*\}} r(\mathbf{X}) = \arg\max_{\mathbf{Y} \in \mathcal{Y} - \{\mathbf{Y}^*\}} r(\mathbf{Y}) \,\middle|\, \mathbf{X}^* = \mathbf{Y}^*\right) P(\mathbf{X}^* = \mathbf{Y}^*).
\end{aligned}
\tag{12}
$$

Since $\mathbf{X}_1, \mathbf{X}_2, \cdots, \mathbf{X}_N$ and $\mathbf{Y}_1, \mathbf{Y}_2, \cdots, \mathbf{Y}_M$ are i.i.d. random variables, we can still decompose the term $P\left(\arg\max\limits_{\mathbf{X} \in \mathcal{X} - \{\mathbf{X}^*\}} r(\mathbf{X}) = \boldsymbol{x} \,\middle|\, \mathbf{X}^* = \mathbf{Y}^*\right)$ as Eq. (9), which means when $N, M \to +\infty$:

$$
P\left(\arg\max_{\mathbf{X} \in \mathcal{X} - \{\mathbf{X}^*\}} r(\mathbf{X}) = \arg\max_{\mathbf{Y} \in \mathcal{Y} - \{\mathbf{Y}^*\}} r(\mathbf{Y}) \,\middle|\, \mathbf{X}^* = \mathbf{Y}^*\right) \to 1.
\tag{13}
$$

Eq. (13) means the elements with the secondary large $r(\boldsymbol{x})$ values are highly likely to be the same. The decomposition in Eq. (12) can be conducted $K$ times for the elements ranked from 1 to $K$, and each decomposed term approaches 1 similar to Eq. (13). So far, we have proved that the probability of $\mathcal{X}$ and $\mathcal{Y}$ having the same top-$K$ subsets measured by $r(\boldsymbol{x})$ approaches to 1 when $N, M \to +\infty$, which is formally written as

$$
P\left(\text{top-}K\left\{r(\mathbf{X}_n) \,|\, 1 \le n \le N\right\} = \text{top-}K\left\{r(\mathbf{Y}_m) \,|\, 1 \le m \le M\right\}\right) \to 1.
\tag{14}
$$

This completes the proof of Proposition 2.1.

## B    MORE EXPERIMENTAL DETAILS

**Model and Training Configurations.**    We mostly follow Brown et al. (2020) to set the model and learning rate configurations, as summarized in Table 6. The 500M, 1.8B, and 4B teacher models are the officially released Qwen-1.5 checkpoints [3] and the 300M teacher model used in Section 3.5 to analysis the effect of teacher LM sizes Is pre-trained on 200B tokens from our pre-training corpus. We train all the LMs with the AdamW (Loshchilov & Hutter, 2019) optimizer, with $\beta_1 = 0.9$, $\beta_2 = 0.98$, and a 0.1 weight decay. We set the batch size to 512 and the max sequence length to 1,024, corresponding to 100K total training steps for roughly 50B tokens in Pre-Train w/o KD, SeqKD, and MINIPLM. For MiniLLM and Vanilla KD, we limit the training steps to align their training computation with Pre-Train w/o KD. Specifically, we assume the computation of a forward and backward pass are $2ND$ and $4ND$, respectively, where $N$ is the model size and $D$ is the number of trained tokens. The training steps of MiniLLM and Vanilla KD are listed in Table 7. We linearly warm up the learning rate for 2K steps and apply cosine learning rate decay until 1/10 of the max values. All experiments are conducted on NVIDIA 40G A100 and NVIDIA 32G V100 GPUs.

**Configurations of KD Across Model Families**    The model configurations of LLaMA3.1 (Dubey et al., 2024) and Mamba (Gu & Dao, 2023) for the cross-family distillation experiments are listed in Table 6. The pre-training data and optimization hyper-parameters are the same as the configurations in our main experiments on Qwen, as listed above.

**Evaluation Details.**    Our downstream datasets for evaluation include Hellaswag (HS; Zellers et al., 2019), LAMBADA (LAM; Paperno et al., 2016), Winograde (Wino; Levesque et al., 2012), OpenbookQA (OBQA; Mihaylov et al., 2018), ARC-Easy/Challange (ARC-e/c; Clark et al., 2018), PIQA (Bisk et al., 2020), SIQA (Sap et al., 2019), and StoryCloze (Story; Mostafazadeh et al., 2016). We apply the LM-Eval-Harness (Gao et al., 2024) [4] framework to conduct zero-shot evaluation. We sample 10K documents from the DCLM (Li et al., 2024a) corpus to construct our test set for language modeling evaluation.

| Family | Size | Vocab. | $d_{\text{model}}$ | $d_{\text{FFN}}$ | $n_{\text{layers}}$ | $n_{\text{head}}$ | $d_{\text{head}}$ | learning rate |
|---|---|---|---|---|---|---|---|---|
| | 104M | 151,936 | 512 | 1,408 | 8 | 8 | 64 | $6 \times 10^{-4}$ |
| | 200M | 151,936 | 768 | 2,112 | 12 | 12 | 64 | $6 \times 10^{-4}$ |
| Qwen | 300M | 151,936 | 768 | 2,112 | 18 | 12 | 64 | $6 \times 10^{-4}$ |
| | 500M | 151,936 | 1,024 | 2,816 | 24 | 16 | 64 | $3 \times 10^{-4}$ |
| | 1.2B | 151,936 | 1,536 | 4,224 | 24 | 16 | 96 | $2.5 \times 10^{-4}$ |
| LLaMA3.1 | 212M | 128,000 | 768 | 3,072 | 12 | 12 | 64 | $6 \times 10^{-4}$ |
| | | Vocab. | $d_{\text{model}}$ | $d_{\text{FFN}}$ | $n_{\text{layers}}$ | conv | rank | learning rate |
| Mamba | 140M | 50,280 | 768 | 1,536 | 24 | 4 | 48 | $3 \times 10^{-3}$ |

Table 6: Model configurations and corresponding learning rates.

| | | Vanilla KD | | | MiniLLM | |
|---|---|---|---|---|---|---|
| Formula | | $\frac{3N_{\text{stu}}}{3N_{\text{stu}}+N_{\text{tch}}}T$ | | | $\frac{3N_{\text{stu}}}{4N_{\text{stu}}+2N_{\text{tch}}}T$ | |
| Student Model Size $N_{\text{stu}}$ | 200M | 500M | 1.2B | 200M | 500M | 1.2B |
| Training Steps | 25K | 45K | 65K | 15K | 30K | 40K |

Table 7: Training steps in Vanilla KD and MiniLLM, which is set to ensure training-time computation to be the same as Pre-Train w/o KD. $N_{\text{stu}}$ and $N_{\text{tch}}$ are model sizes of student and teacher LMs respectively. $N_{\text{tch}} = 1.8B$ in our experiments. $T = 100K$ is the training steps in Pre-Train w/o KD.

## C    DISCUSSION ON RESOURCE CONSUMPTION

In this section, we discuss the asymptotic complexity, actual runtime, and memory usage of different methods. Table C summarizes the offline computation requirements for *Difference Sampling*, which only need to be performed once for any LMs to be distilled from a specific teacher LM. Table C provides the training-time asymptotic complexity, actual runtime, and memory usage for various pre-training methods. For these results, we assume the LMs uses ZeRO-2 (Rajbhandari et al., 2020) and FP16 precision during training to calculate space complexity. The "Actual Time" and "Max Batch Size" metrics are derived from experiments conducted on 8 A100 80GB GPUs. We can see that the inference time of the teacher and reference LMs constitutes the majority of the runtime for *Difference Sampling*. This cost can be further reduced by leveraging a proxy model for sampling, as detailed in Appendix E. Regarding training-time resource consumption, MINIPLM requires the same time and memory as pre-training without KD, whereas Vanilla KD incurs significantly higher runtime and GPU memory usage due to the online inference of the teacher LM.

## D    MORE RESULTS

### D.1    COMPUTATION SCALING CURVES OF MORE SIZES

In Figure 7, we plot the scaling curves of average accuracy on the downstream tasks with respect to the pre-training FLOPs for student LMs with 200M and 1.2B parameters. We can see that MINIPLM saves pre-training computation for both student LM sizes and constantly outperforms Vanilla KD.

### D.2    TEST LOSSES EXTRAPOLATION WITH SCALING LAWS

**Data-Unlimited Setting.**    In Table 2, we follow Hoffmann et al. (2022) to fit the scaling law curves with the test losses on the DCLM corpus. Then, we used the fitted constants to predict the test losses for 1T and 10T training data in Pre-Train w/o KD and MINIPLM, and that for Vanilla KD when it

---

[3] https://huggingface.co/Qwen
[4]

|  | Time Complexity | Space Complexity | Actual Time | Max Batch Size |
|---|---|---|---|---|
| Reference LM Training | $O(6N_{\text{ref}}D_{\text{ref}})$ | $O(3.75N_{\text{ref}})$ | 2.6h | 32 |
| Teacher LM Inference | $O(2N_{\text{tch}}D)$ | $O(2N_{\text{tch}})$ | 68h | 16 |
| Reference LM Inference | $O(2N_{\text{ref}}D)$ | $O(2N_{\text{ref}})$ | 7h | 128 |
| Top-k Sampling | $O(D)$ | - | 10min | - |

Table 8: Asymptotic complexity, actual time and memory use of *Difference Sampling*. $N_{\text{ref}}$ and $N_{\text{tch}}$ are the reference and teacher model sizes. $D$ and $D_{\text{ref}}$ are the sizes of the original corpus and the corpus for training the reference model. "Actual Time" is the running time of each stage. "Max Batch Size" represents the max single GPU batch size, which measures the memory use of each stage.

|  | Time Complexity | Space Complexity | Actual Time | Max Batch Size |
|---|---|---|---|---|
| Pre-Train | $O(6N_{\text{stu}}D)$ | $O(3.75N_{\text{stu}})$ | 68h | 16 |
| Vanilla KD | $O(6N_{\text{stu}}D + 2N_{\text{tch}}D)$ | $O(3.75N_{\text{ref}} + 2N_{\text{tch}})$ | 139h | 8 |
| MiniPLM | $O(6N_{\text{stu}}D)$ | $O(3.75N_{\text{stu}})$ | 68h | 16 |

Table 9: Training-time asymptotic complexity, actual time and memory use of different pre-training methods. $N_{\text{stu}}$ and $N_{\text{tch}}$ are the student and teacher model sizes. $D$ is the size of the pre-training corpus. "Actual Time" is the training time of each method. "Max Batch Size" represents the max single GPU batch size, which measures the memory use of each method. Unlike the setting in Section 3.1, we align the total training steps (token numbers $D$) of the three methods to compare their training-time complexities.

consumes the same training FLOPs. Specifically, we consider the following power law:

$$L(N_m, N_d) = L_{\text{irr}} + \frac{A_m}{N_m^{\alpha_m}} + \frac{A_d}{N_d^{\alpha_d}} \tag{15}$$

where $N_m$ is the model sizes, $N_d$ is the number of trained tokens, and $L_{\text{irr}}$ is the irreducible test loss. Since the required computation $C$ of training on a token of Vanilla KD and other methods are different, we re-write Eq. (15) using the fact that $C \propto N_m N_d$ for a fixed model size:

$$L(C) = L_\infty + \frac{A_c}{C^{\alpha_c}}, \tag{16}$$

where $L_\infty$ and $A_c$ depends on the model size $N_m$, and $\alpha_c = \alpha_d$. In this way, we can fit the loss curves in Figure 4 with Eq. (16). Table 10 includes the fitted values of $L_\infty$, $A_c$, and $\alpha_c$. We also include the total FLOPs of Pre-Train w/o KD and MiniPLM on 1T ($C_{1\text{T}}$) and 10T ($C_{10\text{T}}$) tokens, which Vanilla KD aligns with. The numbers in Table 10 can be computed with Eq. (16) and the constants in Table 10.

**Data-Limited Setting.** In Section 3.4, we extrapolate the loss curve of Pre-Train w/o KD with the Data-Constrained Scaling Law (Muennighoff et al., 2023), which is almost the same as Eq. (15), except that $N_d$ represents the total number of tokens in the pre-training corpus repeated 4 times, as suggested by Muennighoff et al. (2023). Therefore, after $N_d$ is solved by letting the loss of Pre-Train w/o KD equal that of MiniPLM ($\alpha = 0.25$, 4 Eps.), we divide the value by 4, resulting in a training tokens number of 118B. This means 68B extra training tokens are needed for Pre-Train w/o KD to achieve the performance of MiniPLM using 50B training tokens.

### D.3 ABLATION STUDY

**Impact of the Reference Model.** In Figure 8, we show the impact of the reference model size on the performance of LMs trained with MiniPLM. We can see that larger reference models lead to better performance of MiniPLM, but the improvement saturates as the reference LM grows larger.

**Difference Sampling Ratio.** In Figure 9, we plot the impact of the difference sampling ratio on the student LM's performance in the data-unlimited setting. We can see that small sampling ratios result in better zero-shot accuracy on downstream tasks. However, to ensure that the difference-sampled data contains 50B tokens, we need a larger original corpus $\mathcal{D}$. To control the computation overhead, we mainly use $\alpha = 0.5$ in our experiments.

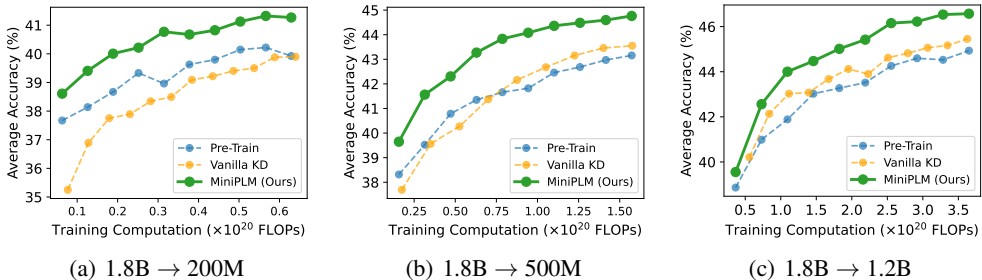

Figure 7: Computation scaling curves of student LMs when trained with Pre-Train w/o KD, Vanilla KD, and MINIPLM. The y-axis represents the average zero-shot accuracy on downstream tasks.

| $N_{\text{stu}}$ | Method | $A_c$ | $\alpha_c$ | $L_\infty$ | $C_{1\text{T}}$ (FLOPs) | $C_{10\text{T}}$ (FLOPs) |
|---|---|---|---|---|---|---|
| 200M | Pre-Train w/o KD | $2.19{\times}10^7$ | 0.41 | 3.30 | | |
| | Vanilla KD | $9.77{\times}10^7$ | 0.44 | 3.34 | $1.26{\times}10^{21}$ | $1.26{\times}10^{22}$ |
| | MINIPLM | $8.56{\times}10^{10}$ | 0.59 | 3.25 | | |
| 500M | Pre-Train w/o KD | $2.73{\times}10^8$ | 0.45 | 3.06 | | |
| | Vanilla KD | $3.14{\times}10^8$ | 0.45 | 3.05 | $3.14{\times}10^{21}$ | $3.14{\times}10^{21}$ |
| | MINIPLM | $6.64{\times}10^9$ | 0.52 | 3.03 | | |
| 1.2B | Pre-Train w/o KD | $1.88{\times}10^8$ | 0.43 | 2.91 | | |
| | Vanilla KD | $1.10{\times}10^{10}$ | 0.52 | 2.90 | $7.30{\times}10^{21}$ | $7.30{\times}10^{21}$ |
| | MINIPLM | $4.29{\times}10^8$ | 0.45 | 2.86 | | |

Table 10: Scaling Law constants in Eq. (16) fitted using the loss curves in Figure 4. $N_{\text{stu}}$ means the student LM size. $C_{1\text{T}}$ and $C_{10\text{T}}$ are the compute spent on processing 1T and 10T tokens in Pre-Train w/o KD and MINIPLM, which Vanilla KD aligns with.

### D.4 Correlations between $\log \frac{p(x)}{p_{\text{REF}}(x)}$ and $\log \frac{p(x)}{q_\theta(x)}$

To examine how replacing $q_\theta$ with $p_{\text{ref}}$ in *Difference Sampling* works, we compute the correlations between $\log \frac{p(x)}{p_{\text{ref}}(x)}$ (using 45M, 104M, and 200M LMs to generate $p_{\text{ref}}$) and $\log \frac{p(x)}{q_\theta(x)}$ (using the 500M student LM to generate $q_\theta$) on a holdout set. Additionally, we calculate the sampling accuracy, which measures how many instances are correctly classified as selected or discarded when using $\log \frac{p(x)}{p_{\text{ref}}(x)}$ and setting the sampling threshold to 0.5, with the selection using $\log \frac{p(x)}{q_\theta(x)}$ serving as the ground truth. As shown in Table D.3, using $p_{\text{ref}}$ achieves strong correlations with using $q_\theta$ and high data sampling accuracy. This supports the rationale for using $p_{\text{ref}}$ in *Difference Sampling* as described in Section 2.2.

| $N_{\text{ref}}$ | Pearson | Spearman | Sampling Acc. |
|---|---|---|---|
| 45M | 0.743 | 0.774 | 80.6 |
| 104M | 0.826 | 0.840 | 87.4 |
| 200M | 0.856 | 0.879 | 89.5 |

Table 11: Correlations between $\log \frac{p(x)}{p_{\text{ref}}(x)}$ and $\log \frac{p(x)}{q_\theta(x)}$. $N_{\text{ref}}$ represents the size of the reference model. $q_\theta$ is generated from the 500M student LM. We report the Pearson/Spearman Correlation and the sampling accuracy (Sampling Acc.) of using $\log \frac{p(x)}{p_{\text{ref}}(x)}$.

## E Difference Sampling with a Proxy Model

**Motivation.** In this section, we show that the offline computational overhead of *Difference Sampling* can be further reduced by conducting teacher and reference LM inference on a small proxy dataset and then transferring the value $r(p, p_{\text{ref}}, \boldsymbol{x})$ to the entire corpus $\mathcal{D}$ with a proxy model. Specifically, we first uniformly sample a proxy subset $\mathcal{D}_{\text{prx}}$ from $\mathcal{D}$, satisfying $|\mathcal{D}_{\text{prx}}| \ll |\mathcal{D}|$. Then, we compute the $r(p, p_{\text{ref}}, \boldsymbol{x})$ value for each instance $\boldsymbol{x}$ in $\mathcal{D}_{\text{prx}}$. Note that since $|\mathcal{D}_{\text{prx}}| \ll |\mathcal{D}|$, the computational

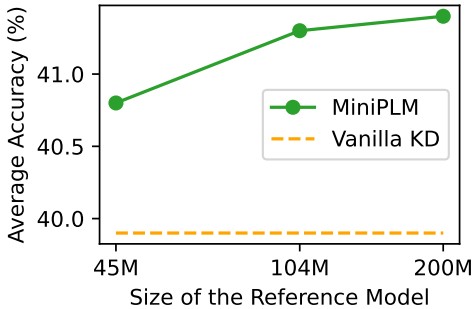 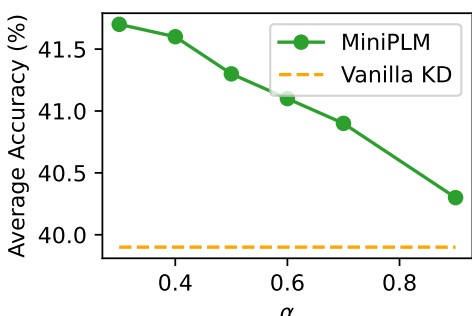

Figure 8: Impact of the reference model size. We use the 1.8B LM as the teacher and the 200M LM as the student. We report the average zero-shot accuracy on the downstream tasks of the LMs trained with MINIPLM and compare it with that of Vanilla KD.

Figure 9: Impact of the difference sampling ratio $\alpha$. We report the average zero-shot accuracy on the downstream tasks of the LMs trained with MINIPLM, using $\alpha \in [0.3, 0.4, 0.5, 0.6, 0.7, 0.9]$ and compare it with that of Vanilla KD.

overhead of teacher LM inference is significantly reduced. After that, we fine-tune a small proxy model on $\mathcal{D}_{\text{prx}}$ to fit the $r(p, p_{\text{ref}}, \boldsymbol{x})$ values, which is used to infer the values for instances from $\mathcal{D}$. Finally, the Top-$K$ operation in Eq. (4) is based on the inferred values.

**Method.** To test this approach, we uniformly sample a $\mathcal{D}_{\text{prx}}$ containing 0.1B tokens from the 50B-token $\mathcal{D}$. Computing $\log \frac{p(\boldsymbol{x})}{p_{\text{ref}}(\boldsymbol{x})}$ values on $\mathcal{D}_{\text{prx}}$ takes only 0.2% computation of that on $\mathcal{D}$. Then, we employ the reference LM as the proxy model and fine-tune it with the following regression loss:

$$\boldsymbol{w}^*, b^*, \boldsymbol{\theta}^*_{\text{ref}} = \arg \min_{\boldsymbol{w}, b, \boldsymbol{\theta}_{\text{ref}}} \frac{1}{|\mathcal{D}_{\text{prx}}|} \sum_{\boldsymbol{x} \in \mathcal{D}_{\text{prx}}} \left[ \boldsymbol{w}^\top \overline{\boldsymbol{h}}(\boldsymbol{x}, \boldsymbol{\theta}_{\text{ref}}) + b - r(p, p_{\text{ref}}, \boldsymbol{x}) \right]^2, \tag{17}$$

where $\overline{\boldsymbol{h}}(\boldsymbol{x}, \boldsymbol{\theta}_{\text{ref}}) \in R^d$ is the average output hidden states of the reference LM, with $d$ representing the model's hidden size and $\boldsymbol{\theta}_{\text{ref}}$ representing the parameters of the reference LM. $\boldsymbol{w} \in \mathbb{R}^d, b \in \mathbb{R}$ are the parameters of a linear head outputting a scalar as the predicted $r(p, p_{\text{ref}}, \boldsymbol{x})$ values, given the average hidden states. The inferred values on $\mathcal{D}$ are given by $\hat{r}(\boldsymbol{x}) = \boldsymbol{w}^{*\top} \overline{\boldsymbol{h}}(\boldsymbol{x}, \boldsymbol{\theta}^*_{\text{ref}}) + b^*$. Since this inference process is based on the reference LM, it still saves computation compared to inference with the teacher LM. The difference-sampled corpus is then obtained by selecting $K = \alpha|\mathcal{D}|$ instances from $\mathcal{D}$ with the highest $\hat{r}(\boldsymbol{x})$ values.

| Method | FLOPs | Acc. |
|---|---|---|
| Vanilla KD | Online | 39.9 |
| MINIPLM | $2 \times 10^{20}$ | **41.3** |
| MINIPLM$_{\text{prx}}$ | $9 \times 10^{18}$ | 40.9 |

Table 12: Offline FLOPs and average accuracy (Acc.) on downstream tasks of MINIPLM using a proxy model, compared with that of standard MINIPLM and Vanilla KD.

**Results.** We term this method as MINIPLM$_{\text{prx}}$ and compare its performance with Vanilla KD and MINIPLM in Table 12. We can see that MINIPLM$_{\text{prx}}$ requires much less inference computation compared to standard MINIPLM while still maintaining substantial improvement over Vanilla KD.

## F  CASE STUDY

We present a case study in Table 13, 14, and 15 to show the instances corresponding to the three parts of data distribution shown in Figure 3(b). The explanation of the three parts are list as follows:

- $p(\boldsymbol{x}) \gtrsim p_{\text{ref}}(\boldsymbol{x})$: As shown in Table 13, the instances whose $\log \frac{p(\boldsymbol{x})}{p_{\text{ref}}(\boldsymbol{x})}$ value is close to 0 is discarded (Instance #1 and #2). These instances are full of repeated patterns, which is easily learned by both the small and large LMs. Instance #3 is a dialog from a story, where the teacher LM learns slightly well than the student LM, which is retained for the student LM to

learn basic language skills. In this way, these easy and common patterns are down-sampled but not fully discarded.

- $p(\boldsymbol{x}) \gg p_{\text{ref}}(\boldsymbol{x})$: As shown in Table 14, the examples where $p(\boldsymbol{x})$ is much larger than $p_{\text{ref}}(\boldsymbol{x})$ is retained. These examples include long and knowledge-intensive documents (Instance #1), in-context learning examples (Instance #2), and high-quality codes (Instance #3). These examples reveals the key large LM's advantage over the small LM, which should be up-sampled for the student LM to learn.

- $p(\boldsymbol{x}) \lesssim p_{\text{ref}}(\boldsymbol{x})$: As shown in Table 15, the instances where the larger LM learns worse than the small LM are all discarded. We can see that Instance #1 and Instance #2 are irregular noises, which will be harmful to the training of LMs. The small reference LM easily over-fits their surface patterns like symbols and numbers, while a more powerful large LM better recognizes their unnatural features. Instance #3 is a data collection in an academic paper, but lack of contexts, which is useless for LMs to learn the dependency in contexts. Note that these examples only accounts for 6.3% of the pre-training corpus before *Difference Sampling*, which is a quite small proportion.

| $p(\boldsymbol{x}) \gtrsim p_{\text{ref}}(\boldsymbol{x})$: Easy and common instances | | | | |
|---|---|---|---|---|
| Instance #1 | $-\log p(\boldsymbol{x}) = 1.24$ | $-\log p_{\text{ref}}(\boldsymbol{x}) = 1.28$ | $\log \frac{p(\boldsymbol{x})}{p_{\text{ref}}(\boldsymbol{x})} = 0.04$ | Discarded |

```
<node id='-659' action='modify' visible='true' lat=
'0.05467069248579575' lon='0.014892969670168166' />
<node id='-657' action='modify' visible='true' lat=
'0.05243834625645345' lon='0.023237696125627347' />
<node id='-655' action='modify' visible='true' lat=
'0.04940873338995851' lon='0.03152927145717611' />
<node id='-653' action='modify' visible='true' lat=
'0.04786735135170292' lon='0.040405509151806455' />
<node id='-651' action='modify' visible='true' lat=
'0.04733584029593642' lon='0.04513595918070425' />
```

| Instance #2 | $-\log p(\boldsymbol{x}) = 0.44$ | $-\log p_{\text{ref}}(\boldsymbol{x}) = 0.51$ | $\log \frac{p(\boldsymbol{x})}{p_{\text{ref}}(\boldsymbol{x})} = 0.07$ | Discarded |

```
{\sum\limits_{j = 1}^{n - 1}\operatorname{size}\left(
{\mathcal{X},j} \right) \cdot \operatorname{size}
\left( {\mathcal{Z},n - j} \right)\quad} &
{\mathcal{I} = \mathcal{M}_{1},} \\
{\sum\limits_{j = 1}^{n - 1}\operatorname{size}\left(
{\mathcal{X},j} \right) \cdot \operatorname{size}
\left( {\mathcal{X},n - j} \right)\quad} &
{\mathcal{I} = \mathcal{M}_{2},} \\
{\sum\limits_{j = 1}^{n - 1}\operatorname{size}\left(
{\mathcal{X},j} \right) \cdot \operatorname{size}
\left( {\mathcal{A},n - j} \right)\quad} &
{\mathcal{I} = \mathcal{M}_{3},} \\
{\sum\limits_{j = 1}^{n - 1}\operatorname{size}\left(
{\mathcal{A},j} \right) \cdot \operatorname{size}
\left( {\mathcal{Z},n - j} \right)\quad} &
{\mathcal{I} = \mathcal{M}_{4},} \\
```

| Instance #3 | $-\log p(\boldsymbol{x}) = 2.83$ | $-\log p_{\text{ref}}(\boldsymbol{x}) = 3.86$ | $\log \frac{p(\boldsymbol{x})}{p_{\text{ref}}(\boldsymbol{x})} = 1.02$ | Selected |

```
"I felt bad for Coach Rod to have to deal with it,"
said senior tight end Mike Massey, a St.  Ignatius
grad.  "But for the players it was a non-issue.  We
didn't even talk about it."

But some other people in college football did.

"[Boren] was a legacy guy and when he makes comments
like [that], that's like getting kicked square in the
shorts when you're Rich Rodriguez," ESPN analyst and
former OSU quarterback Kirk Herbstreit said Thursday.
"With all the other things that are happening, when
you get that comment from a legacy guy, a guy whose
dad started three or four years for Bo, all of a
sudden everybody's tentacles go up a little bit.
```

Table 13: Easy and common instances down-sampled by *Difference Sampling*. Instance #1 and #2: HTML and LaTeX code data that contain repeated patterns, which is easy to fit by both the reference and teacher LM. Instance #3: Dialogues from a story, which is relative easy for LMs but are still selected by *Difference Sampling* to help student LMs learn basic language skills.

| $p(\boldsymbol{x}) \gg p_{\text{ref}}(\boldsymbol{x})$: Hard and valuable instances | | | | |
|---|---|---|---|---|
| Instance #1 | $-\log p(\boldsymbol{x}) = 1.26$ | $-\log p_{\text{ref}}(\boldsymbol{x}) = 4.20$ | $\log \frac{p(\boldsymbol{x})}{p_{\text{ref}}(\boldsymbol{x})} = 2.94$ | Selected |

```
Legal along with Environmental Responsibility!
Dumpster rentals in the user side may seem as
fundamental as placing a phone, having a dumpster sent
and hurling all your disposals inside to be carted
away.  Nonetheless, there are legal issues attached
to appropriate disposal connected with certain products
which tie up into environmental issues.  The 10 Yard
Dumpster For Rent in Pocahontas customer or perhaps
demolition purchaser should be informed about these
issues by means of careful screening so as to reduce a
firm's liability which inturn keeps a firm's overhead
all the way down and makes for prompt fall off, pick up
along with disposal of the dumpster and it's articles.
```

| | | | | |
|---|---|---|---|---|
| Instance #2 | $-\log p(\boldsymbol{x}) = 2.36$ | $-\log p_{\text{ref}}(\boldsymbol{x}) = 5.59$ | $\log \frac{p(\boldsymbol{x})}{p_{\text{ref}}(\boldsymbol{x})} = 3.23$ | Selected |

```
有利 you3li4 yǒulì advantageous; beneficial
谨慎 jin3shen4 jǐnshèn cautious; prudent
甲 jia3 jiǎ one; armor (1st Heavenly Stem)
犹豫 you2yu4 yóuyù hesitate; hesitant; undecided
从此 cong2ci3 cóngcǐ from now on; since then
企业 qi3ye4 qǐyè company; business; firm
下载 xia4zai3 xiàzǎi to download
狮子 shi1zi5 shīzi lion
青少年 qing1shao4nian2 qīngshàonián teenager
```

| | | | | |
|---|---|---|---|---|
| Instance #3 | $-\log p(\boldsymbol{x}) = 0.16$ | $-\log p_{\text{ref}}(\boldsymbol{x}) = 2.73$ | $\log \frac{p(\boldsymbol{x})}{p_{\text{ref}}(\boldsymbol{x})} = 2.56$ | Selected |

```
function WritableState(options, stream) {
  var Duplex = require('./_stream_duplex');
  options = options || {};
  // the point at which write() starts returning false
  // Note:  0 is a valid value, means that we always
  return false if
  // the entire buffer is not flushed immediately
  on write()
  var hwm = options.highWaterMark;
  var defaultHwm = options.objectMode?16:16*1024;
  this.highWaterMark = (hwm || hwm === 0) ?  hwm :
  defaultHwm;

  // object stream flag to indicate whether or not
  this stream
  // contains buffers or objects.
  this.objectMode = !!options.objectMode;
  ...
}
```

Table 14: Hard and valuable instances up-sampled by *Difference Sampling*. Instance #1: High-quality long documents containing versatile world knowledge. Instance #2: instance that contains translation tasks, presented in an in-context learning form. Instance #3: High-quality code data with detailed comments.

| $p(\boldsymbol{x}) \lesssim p_{\text{ref}}(\boldsymbol{x})$: Noisy and harmful instances | | | |
|---|---|---|---|
| Instance #1 | $-\log p(\boldsymbol{x}) = 9.50$ $\bigm|$ $-\log p_{\text{ref}}(\boldsymbol{x}) = 6.60$ $\bigm|$ $\log \frac{p(\boldsymbol{x})}{p_{\text{ref}}(\boldsymbol{x})} = -2.90$ $\bigm|$ Discarded | | |

```
]{}
**********************************************

.  ...

...   .
```

| Instance #2 | $-\log p(\boldsymbol{x}) = 1.01$ $\bigm|$ $-\log p_{\text{ref}}(\boldsymbol{x}) = 0.90$ $\bigm|$ $\log \frac{p(\boldsymbol{x})}{p_{\text{ref}}(\boldsymbol{x})} = -0.11$ $\bigm|$ Discarded |

```
(91.00,60.00) (1.00,35.00) (2.00,35.00)[(1,0)[13.00]{}]
{} (16.00,35.00) (16.70,34.30)[(1,-1)[8.60]{}]{}
(26.00,25.00) (26.00,11.00) (26.00,12.00)[(0,1)[12.00]
{}]{} (36.00,35.00) (26.00,45.00) (16.70,35.70)
(35.30,34.30)[(-1,-1)[8.60]{}]{} (56.00,35.00)
(66.00,25.00) (76.00,35.00) (66.00,45.00) (66.00,46.00)
```

| Instance #3 | $-\log p(\boldsymbol{x}) = 2.53$ $\bigm|$ $-\log p_{\text{ref}}(\boldsymbol{x}) = 0.26$ $\bigm|$ $\log \frac{p(\boldsymbol{x})}{p_{\text{ref}}(\boldsymbol{x})} = -2.26$ $\bigm|$ Discarded |

```
*Z* = 8
---------------- -------------------------

Data collection #tablewrapdatacollectionlong
===============

-----------------------------------------
Bruker APEXII area-detector diffractometer
Radiation source:  fine-focus sealed tube
graphite
p and w scans
Absorption correction:  multi-scan
(*SADABS*; Sheldrick, 2004)
*T*m̃in≅ 0.970, *T*m̃ax≅ 0.981
20408 measured reflections
-----------------------------------------
```

Table 15: Noisy and harmful instances discarded by *Difference Sampling*. Instance #1: irregular symbols. Both the reference LM and the teacher LM are hard to fit. Instance #2: a string primarily made of meaningless numbers. Both the reference LM and the teacher LM are easy to fit, by predicting numbers. Instance #3: data collection in scientific paper, but lack of contexts. The reference LM fits the patterns, which the teacher LM finds useless.

