# OpenReview forum: "MiniPLM: Knowledge Distillation for Pre-training Language Models"
_ICLR.cc/2025/Conference — ICLR 2025 Poster_

### Official Review · Reviewer_rC3o · 2024-10-29

**Soundness:** 3
**Presentation:** 3
**Contribution:** 2
**Rating:** 6
**Confidence:** 3

**Summary:**

The paper proposes a method to train a small PLM using knowledge distillation from large PLM and in an offline manner instead of the existing online methods. The methods first trains a small LM from scratch on small subset of the training corpus and then a large PLM is used to perform difference sampling that samples from the large corpus the difficult samples reducing the chances of learning the patterns only from the easy portion of the dataset.

**Strengths:**

1. The paper uses knowledge distillation to obtain a better subset of samples for training a small PLM which is different from existing methods.

2. The paper is well written and easy to follow.

3. The paper has good experimental results as compared to the existing methods.

4. The problem taken up is an important problem in pre-training the language models.

**Weaknesses:**

1. The paper proposes to choose a subset of training dataset that can provide better performance of the small model using knowledge distillation but the title of the paper is ``Knowledge distillation for pre-training language models", here knowledge distillation is not being used for pretraining instead it is being used for a better subset selection from a huge corpus of data that can in turn have better pre-training of the PLMs, this makes it quite misleading.

2. In line 238 paper says that when the log term is less than zero, those instances are harmful, however, this is not true as explained in papers such as [1] where an easy sample can be predicted correctly by a smaller model while it can get wrong on a larger model due to extraction of overly distracted features. This phenomenon is termed as `overthinking' of PLMs.

3. I am not very clear how $p_{ref}$ can mimic the behavior of $q_{\theta}$, even these models can have certain differences and how MINIplm performs better than MINIllm even when MINIllm directly uses the $q_{\theta}$ distribution?

4. The method is computationally very heavy as we need three LLM model: one large model from which distillation is being performed, a smaller version trained from scratch and then training the distilled model using the existing two, which makes it complex.

**Questions:**

See weaknesses.

---

> ### Author Response · Authors · 2024-11-19
> **About the Reference [1] in Your Comments**
>
> Dear Reviewer,
>
> Thank you for your detailed and thoughtful comments! We are currently working on our response and noticed that the reference [1] mentioned in your comments appears to be missing. We believe this reference would be highly valuable for our response and for further improving our paper. We would greatly appreciate it if the reviewer could kindly provide the reference.
>
> Thanks,
>
> Authors of Submission3739

---

> ### Author Response · Authors · 2024-11-22
> **Response to Reviewer rC3o (part 1/2)**
>
> We sincerely thank the reviewer for recognizing the novelty of our approach, the clarity of our writing, the strong experimental results, and the importance of the problem we address.
>
> ## How knowledge distillation is used (weakness #1)
>
> Knowledge distillation is used for pre-training in MiniPLM because, as explained in Section 2.1, the data sampling process in MiniPLM is **essentially derived from minimizing the reverse KLD between the student and teacher LMs, a widely used objective in KD for training LMs [1,2,3]**. While effective, directly minimizing reverse KLD is challenging to apply during the pre-training stage due to its high computational overhead for optimization. To address this issue, we propose MiniPLM based on *Difference Sampling*, which is derived from the optimization of reverse KLD with approximations supported by theory (Proposition 2.1) and empirical results (Appendix D.4). **MiniPLM offers a more effective, efficient, and flexible solution for KD in pre-training LMs** compared to existing KD methods [4,5] which are primarily used for fine-tuning. This motivation underpins the title of our paper: “Knowledge Distillation for Pre-training Language Models.”
>
> [1] MiniLLM: Knowledge Distillation of Large Language Models. 2024. In ICLR.
>
> [2] On-policy distillation of language models: Learning from self-generated mistakes. 2024. In ICLR.
>
> [3] DistiLLM: Towards Streamlined Distillation for Large Language Models. 2024. In ICML.
>
> [4] Vicuna: An open-source chatbot impressing gpt-4 with 90% chatgpt quality. 2023.
>
> [5] Compact Language Models via Pruning and Knowledge Distillation. 2024. In NeurIPS.
>
> ## The instances with negative scores (weakness #2)
>
> We thank the reviewer for pointing out the “overthinking” phenomenon of PLMs. During our response, we noticed that the reference [1] mentioned in your comments appears to be missing. We believe this reference would be highly valuable for our response and for further improving our paper. **Without the reference, it is hard for us to determine how the instances causing the "overthinking" phenomenon look like, which is critical for us to explore how these instances affect the pre-training of LMs.** We would greatly appreciate it if the reviewer could kindly provide the reference.
>
> ## How $p_{\text{ref}}$ can mimic the behavior of $q_{\theta}$ (weakness #2)
>
> We use $p_{\text{ref}}$ to approximate $q_{\theta}$ for reasons of computational efficiency. While using $q_\theta$ directly might yield better performance in MiniPLM, as suggested by Figure 8, where larger $p_{\text{ref}}$ models lead to improved results, we find that smaller $p_{\text{ref}}$ models already provide strong performance.
>
> To further support this claim, we compute the correlations between $\log \frac{p(x)}{p_{\text{ref}}(x)}$ (using 45M, 104M, and 200M models for $p_{\text{ref}}$) and $\log \frac{p(x)}{q_{\theta}(x)}$ (using the 500M student model for $q_{\theta}$) on a holdout set of the Pile. Additionally, we calculate the sampling accuracy, which measures how many instances are correctly classified as selected or discarded when using $\log \frac{p(x)}{p_{\text{ref}}(x)}$ with a sampling threshold of 0.5. The ground truth is determined by sampling based on $\log \frac{p(x)}{q_{\theta}(x)}$.  As the following table shows, using $p_{\text{ref}}$ achieves high correlations with $q_{\theta}$ and high data sampling accuracy. We have added these results to Appendix D.4 in the revised paper.
>
> | Reference Model Size | Pearson Correlation | Spearman Correlation | Sampling Accuracy (%) |
> | --- | --- | --- | --- |
> | 45M | 0.743 | 0.774 | 80.6 |
> | 104M | 0.826 | 0.840 | 87.4 |
> | 200M | 0.856 | 0.879 | 89.5 |
>
> Similar strategies, where small models approximate probability differences for larger models, have been successfully employed in quite a few prior works [1,2,3,4].
>
> [1] Selection via Proxy: Efficient Data Selection for Deep Learning. 2020. In ICLR.
>
> [2] DoReMi: Optimizing Data Mixtures Speeds Up Language Model Pre-training. 2023. In NeurIPS.
>
> [3] Perplexed by Perplexity: Perplexity-Based Data Pruning With Small Reference Models. 2024. arxiv pre-print.
>
> [4] Irreducible Curriculum for Language Model Pre-training. 2024. arxiv pre-print.

---

> > ### Author Response · Authors · 2024-11-22
> > **Response to Reviewer rC3o (part 2/2)**
> >
> > ## How MiniPLM performs better than MiniLLM
> >
> > MiniPLM outperforms MiniLLM because the two methods are compared under equivalent training computation, which is a more appropriate control for pre-training language models [1,2]. MiniLLM incurs significant computational overhead as it requires sentence generation from the student distribution and online teacher model inference during training. This overhead arises from directly using $q_{\theta}$, which requires continuously updating the generated sentences as $\theta$ evolves during training. This highlights the necessity of using $p_{\text{ref}}$ for efficiency, as demonstrated in MiniPLM.
> >
> > [1] No Train No Gain: Revisiting Efficient Training Algorithms For Transformer-based Language Models. 2023. In NeurIPS.
> >
> > [2] Training Compute-Optimal Large Language Models. 2022. In NeurIPS.
> >
> > ## The complexity of MiniPLM
> >
> > - Regarding "The method is computationally very heavy as we need three LLM models": The use of teacher and student models is a fundamental setup in knowledge distillation and typically involves student training and teacher inference. As shown in Tables 8 and 9, the training and inference of the reference LM (2.6h + 7h = 9.6h) account for only **7% of the computation** in a conventional KD setting (68h + 68h = 136h), making this overhead negligible.
> > - Regarding the computational complexity, MiniPLM adopts offline teacher LM inference, which is conducted **only once** to distill multiple student LMs from a specific teacher LM.
> >     - Compared to Pre-tran w/o KD, no extra training computation is introduced for each student LM. Moreover, as demonstrated in Appendix E, the offline computation for the teacher and reference LMs can be **reduced by 95%** using a proxy model in Difference Sampling, with only minimal performance tradeoff.
> >     - Compared to Vanilla KD, the offline nature makes MiniPLM quite efficient. This is because Vanilla KD requires online teacher LM inference during the pre-training of every student LM to be distilled from the teacher LM. This extra computation is quite heavy because the teacher LM is typically much larger than the student LM.
> > - Regarding the implementation complexity, the MiniPLM pipeline is straightforward and fully compatible with recent training and inference frameworks. The computation of $\log p(x)$ and $\log p_{\text{ref}}(x)$ is well-supported by open-source libraries [1,2]. The top-k sampling can be implemented with only ~60 lines of code to run on CPUs. More importantly, once $\log \frac{p(x)}{p_{\text{ref}}(x)}$ values are computed for a given corpus, the teacher LM’s knowledge can be distilled into any student LMs in future development, **even without the access to the teacher or reference models.**
> >     - Compared to Pre-Train w/o KD, the pre-training framework of the student and reference LMs is nearly identical, with only a minor adjustment to the data loading path. This makes the implementation complexity of using MiniPLM almost the same as Pre-Train w/o KD.
> >     - Compared to Vanilla KD, the implementation of MiniPLM is far simpler, especially for efficient purposes. This is because Vanilla KD requires keeping the teacher LM’s parameters in GPU memory for online inference, necessitating significant engineering efforts to optimize memory usage and avoid out-of-memory (OOM) issues.
> >
> >     [1] DeepSpeed Inference: Enabling Efficient Inference of Transformer Models at Unprecedented Scale. 2022. arxiv pre-print.
> >
> >     [2] Transformers: State-of-the-Art Natural Language Processing. 2020. In ACL.

---

> ### Author Response · Authors · 2024-11-25
> **We hope the reviewer find our response helpful**
>
> We highly appreciate the reviewer’s insightful feedback, which clearly helped us improve our paper.
>
> In our response, we have addressed key concerns, including how knowledge distillation is applied during pre-training, how effectively $p_{\text{ref}}$ mimics the behavior of $q_{\theta}$, why MiniPLM outperforms MiniLLM, and the relatively low complexity of MiniPLM compared to other pre-training or knowledge distillation methods.
>
> With the deadline approaching, we sincerely hope the reviewer find our response useful and update the scores if the concerns have been resolved. We also hope the reviewer could kindly provided the reference [1] in the comments for us to learn more about the “overthinking” problem of PLMs and its relationship to our methods. We remain open to further discussions and are happy to address any additional questions.

---

> > ### Comment · Reviewer_rC3o · 2024-11-25
> > **Rebuttal acknowledgement**
> >
> > Thanks for the rebuttal.
> >
> > The paper that I was referring to is :
> >
> > https://arxiv.org/pdf/2006.04152
> >
> > Please have a look and share your thoughts.
> >
> > Thanks

---

> > > ### Author Response · Authors · 2024-11-26
> > >
> > > We thank the reviewer for providing the reference link of [1]. However, we argue that there is insufficient evidence to suggest that the “overthinking” problem described in [1] applies to our setting, as the experimental setups differ significantly from ours in terms of both model architecture, data distribution, and the selection of small LMs.
> > >
> > > - **Model Architecture**: The “overthinking” problem is primarily observed in ALBERT, an encoder-only model with weight-sharing across layers. In contrast, our work focuses on decoder-only models with no weight-sharing. In fact, the experiments in [1] suggest that the “overthinking” phenomenon is related to the weight-sharing design, as it is not observed in BERT, a model without weight-sharing just like the architectures in our studies.
> > > - **Data Distribution**: [1] investigates the “overthinking” problem in models fine-tuned on text classification/regression datasets, whereas our work focuses on pre-training corpora, which have entirely different data distributions. The datasets used in [1] typically contain short and downstream-task-specific instances, where the LM’s loss is computed on the final classification label. However, the instances from our pre-training corpora are mostly web-crawled long documents, where the loss is computed on the entire sequence with 1024 tokens.
> > > - **The selection of small LMs**: The small LMs in [1] are constructed by extracting some early layers from the large LM while the reference LMs in our paper is trained independently from the large LMs.
> > >
> > > Moreover, **we did not find evidence in [1] to support the claim that samples correctly predicted by smaller models but incorrectly by larger models are “easy” (as mentioned in the reviewer’s comments) or useful for fine-tuning PLMs.** On the contrary, **by analyzing the pre-training corpus where $p_{\text{ref}}(x) > p(x)$, we indeed observed that these data points are mostly meaningless strings, which is consistent with the claims in our paper** . As discussed in Appendix F, smaller models tend to capture surface-level features, assigning high probabilities to these instances, while larger models prioritize deeper, context-level coherence, resulting in low probabilities for such meaningless strings. Previous studies [Li et al., 2024; Weber et al., 2024] have shown that such instances are harmful to pre-training language models.
> > >
> > > Additionally, similar phenomena have been observed in prior literature with experimental setups closer to ours (non-weight-sharing models and pre-training corpora) [Xia et al., 2022]. Xia et al., (2022) found that samples where smaller LMs assign higher probabilities than larger models often exhibit grammatically correct but semantically unnatural sentences that lack grounding in real-world contexts.
> > >
> > > Based on the significant differences between the settings in [1] and our work, our empirical observations, and evidence from prior literature, we firmly stand by our conclusion that examples with $\log \frac{p(x)}{p_{\text{ref}}(x)} > 0$ are harmful for pre-training. That said, we agree that understanding how the “overthinking” problem manifests in recent LMs is a promising avenue for future exploration, and we sincerely thank the reviewer for highlighting this direction.
> > >
> > > [Xia et al., 2022] Training Trajectories of Language Models Across Scales. In ACL2023.
> > >
> > > [Li et al., 2024] DataComp-LM: In search of the next generation of training sets for language models. arxiv pre-print.
> > >
> > > [Weber et al., 2024] RedPajama: an Open Dataset for Training Large Language Models. arxiv pre-print.

---

> > > > ### Comment · Reviewer_rC3o · 2024-11-26
> > > > **Response**
> > > >
> > > > I do not agree with the authors that the overthinking phenomenon is not there in the BERT model as papers such as [2] highlight that. However, I agree with the authors that the impact might be negligible in their case. Increasing score from 5 --> 6.
> > > >
> > > > [2] https://aclanthology.org/2024.findings-acl.101/

---

### Official Review · Reviewer_Rr1c · 2024-11-03

**Soundness:** 3
**Presentation:** 4
**Contribution:** 3
**Rating:** 6
**Confidence:** 3

**Summary:**

This paper introduces MINIPLM, a knowledge distillation framework for pre-training language models. The key innovation is Difference Sampling, which refines the training data distribution by comparing a large teacher model with a small reference model. The framework: (1) performs offline teacher inference to avoid online KD computation overhead, (2) maintains data difficulty through reference model comparison, and (3) enables cross-family model distillation. Experiments on models from 200M to 1.2B parameters demonstrate improvements in downstream performance, language modeling capability, and computational efficiency.

**Strengths:**

- Well-motivated solution approach
- Elegant Difference Sampling design
- Efficient offline computation strategy
- Comprehensive experiments across model scales

**Weaknesses:**

- Limited analysis of finite-sample scenarios
- Teacher model size limited to 1.8B
- Limited cross-family experiments
- Limited exploration of alternative sampling strategies
- Reference model selection criteria not fully justified

**Questions:**

- Can you provide theoretical justification for using log(p(x)/p_ref(x)) as the sampling metric?
- How does the method's effectiveness scale with model size differences?
- How would the method perform with larger teacher models (>10B)?
- What modifications are needed for very large scale deployment?

---

> ### Author Response · Authors · 2024-11-22
> **Response to Reviewer Rr1c (part 1/2)**
>
> We thank the reviewer for acknowledging our well-motivated approach, the elegant design of *Difference Sampling*, the efficient offline computation strategy, and the comprehensive experiments conducted across model scales.
>
> ## Analysis of finite-sample scenarios (weakness #1)
>
> In Section 3.4, we explore the performance of MiniPLM in scenarios where the number of original pre-training tokens (before applying *Difference Sampling*) is limited to 50B. The results demonstrate that MiniPLM significantly improves data utilization under finite-sample conditions.
>
> ## Larger teacher models (weakness #2 and question #3)
>
> We run an additional experiment using the 4B Qwen-1.5 model as the teacher and the 500M model as the student. The results confirm that MiniPLM continues to perform well with a 4B teacher LM.
>
> |                  | HS   | LAM  | Wino | OBQA | ARC-e | ARC-c | PIQA | SIQA | Story | Avg  |
> | ---------------- | ---- | ---- | ---- | ---- | ----- | ----- | ---- | ---- | ----- | ---- |
> | Pre-train w/o KD | 35.8 | 40.1 | 51.0 | 30.2 | 41.7  | 24.4  | 65.4 | 38.2 | 61.4  | 43.2 |
> | Vanilla KD       | 36.8 | 39.9 | 52.0 | 29.2 | 44.3  | 24.1  | 65.3 | 37.8 | 61.3  | 43.4 |
> | MiniPLM          | 38.9 | 42.1 | 52.1 | 31.0 | 44.4  | 24.8  | 66.4 | 39.7 | 62.6  | 44.7 |
>
> Following the trend in Figure 6, the 10B model size may not be optimal for training the student LMs with 200M, 500M, and 1.2B parameters, as the large model size gap can hinder effective knowledge transfer. This phenomenon is consistent with findings in prior knowledge distillation studies [1,2]. We plan to explore distilling the 10B model to larger student LMs (e.g., 4B and 7B) or incorporating the teacher assistants [1] to bridge the model size gap in future work.
>
> [1] Improved Knowledge Distillation via Teacher Assistant. 2020. In AAAI.
>
> [2] MiniLM: Deep Self-Attention Distillation for Task-Agnostic Compression of Pre-Trained Transformers. 2020. In ICML.
>
> ## The cross-family experiments (weakness #3)
>
> As discussed in Section 3.3, we distill the knowledge from the Qwen model to Llama and Mamba models, which covers the most recently used architectures (transformer and state-space models) for language model development. Additional details of these experiments have been included in Appendix B.
>
> ## Alternative sampling strategies  (weakness #4)
>
> We compare *Difference Sampling* with two alternative sampling strategies: sampling based on (1) large model’s preference: $\text{topk}\left[\log p(x)|x\in D\right]$ and (2) the sample difficulty reflected by the reference model: $\text{topk}\left[-\log p_{\text{ref}}(x)|x\in D\right]$.
>
> |                           | Avg. Performance |
> | ------------------------- | ---------------- |
> | Conventional              | 39.9             |
> | Sample difficulty         | 40.2             |
> | Teacher preference        | 37.6             |
> | Difference Sampling(Ours) | **41.3**         |
>
> The results show that considering both the signals of sample difficulty and teacher’s preference, as in *Difference Sampling*, is essential to achieve high performance.
>
> ## The selection criteria of the reference model (weakness #5)
>
> We provide empirical results on the choice of reference model in Figure 8. Basically, the closer the reference model size is to that of the student model, the better the performance. Notably, a reference model with **1/5 of the parameters** of the student model achieves strong results, outperforming Vanilla KD. Users can adjust the reference model size to balance computational cost and final performance based on their specific requirements.
>
> ## Theoretical justification for using $\log \frac{p(x)}{p_{\text{ref}}(x)}$ (question #1)
> Theoretically, the metric $\log \frac{p(x)}{p_\text{ref}(x)}$ approximates to minimizing the reverse KLD between the student and teacher LMs, as described in Eq. (1). This is because minimizing $E_{x\sim q_{\theta}}\log \frac{q_{\theta}(x)}{p(x)}$, can be effectively achieved using the *Best-of-N* algorithm, which selects examples with high $\log \frac{p(x)}{q_{\theta}(x)}$scores from instances sampled by $q_{\theta}$. The $q_{\theta}$-sampled instances can be replaced with the original pre-training corpus, which is theoretically supported by Proposition 2.1 when the dataset size is sufficiently large. To improve efficiency, $q_{\theta}$ in $\log \frac{p(x)}{q_{\theta}(x)}$ is approximated with $p_{\text{ref}}$ , leading to the final metric $\log \frac{p(x)}{p_{\text{ref}}(x)}$.

---

> > ### Author Response · Authors · 2024-11-22
> > **Response to Reviewer Rr1c (part 2/2)**
> >
> > ## Effectiveness for different model size gap (question #2)
> >
> > As shown in Table 6, we compare MiniPLM to baseline methods across a range of model size differences, where the ratio $\frac{\text{teacher model parameters}}{\text{student model parameters}}$ varies from 1.5x to 20x. The performance of MiniPLM follows a first-increase-then-decrease trend when the model size difference scales up, consistent with observations from prior knowledge distillation literature [1,2]. Notably, for larger model size differences, MiniPLM demonstrates even greater improvements over Vanilla KD under the same training computation budget, underscoring its robustness and scalability.
> >
> > [1] Improved Knowledge Distillation via Teacher Assistant. 2020. In AAAI.
> >
> > [2] MiniLM: Deep Self-Attention Distillation for Task-Agnostic Compression of Pre-Trained Transformers. 2020. In ICML.
> >
> > ## Large-scale deployment  (question #4)
> >
> > There are not many modifications required for large-scale deployment. The inference framework of the teacher and reference LMs may need optimization for multi-node computation, which is well supported by many open-source libraries [1,2]. The data sampling process is entirely CPU-based and does not require specific adjustments. The final pre-training stage may need some optimization for high-parallel computing, which are readily available in modern open-source codebases [2,3].
> >
> > [1] DeepSpeed Inference: Enabling Efficient Inference of Transformer Models at Unprecedented Scale. 2022. arxiv pre-print.
> >
> > [2] Megatron-LM: Training Multi-Billion Parameter Language Models Using Model Parallelism. 2019. arxiv pre-print.
> >
> > [3] ZeRO: Memory Optimizations Toward Training Trillion Parameter Models. 2020. arxiv pre-print.

---

> > > ### Comment · Reviewer_Rr1c · 2024-11-25
> > >
> > > Thank you for sovling my concerns. From both theoretical and experimental perspectives, this is a good paper and I support its acceptance.

---

### Official Review · Reviewer_25K6 · 2024-11-04

**Soundness:** 3
**Presentation:** 3
**Contribution:** 3
**Rating:** 6
**Confidence:** 3

**Summary:**

MiniPLM introduces a novel approach to knowledge distillation (KD) for pre-training smaller language models that addresses key challenges in efficiency, flexibility, and effectiveness. The core innovation is "Difference Sampling" - a method that intelligently refines the pre-training data distribution by comparing probabilities between a large teacher model and a small reference model. This enables efficient offline KD by selecting training instances where the teacher model shows strong confidence but the reference model struggles, indicating valuable knowledge that smaller models need to learn. Notably, MiniPLM achieves 2.2x computational savings while maintaining performance comparable to traditional KD approaches.

**Strengths:**

- This paper Introduces "Difference Sampling" as a novel offline KD method that fundamentally changes how knowledge is transferred.
- This paper re-computes teacher model probabilities just once, requiring only 200MB storage for 50B tokens vs 30PB for traditional methods.
- This paper shows clear computational efficiency gains (2.2x reduction for equivalent performance).
- This paper provides solid theoretical justification through Proposition 2.1 and detailed analysis of the sampling approach

**Weaknesses:**

- First, the absence of teacher model (1.8B) performance metrics in Table 1 makes it difficult to gauge the effectiveness of knowledge transfer - readers cannot assess how much of the teacher's capabilities are preserved in the student models.
- The paper's experimental scope is rather constrained. By focusing primarily on a 1.8B teacher model, it misses the opportunity to demonstrate effectiveness with larger, more modern models. The experiments would be more convincing if they demonstrated success with larger model gaps, such as distilling from Llama 3.1 8B to 1B on Table 1 setting.
-  While the paper introduces "Difference Sampling" as a key innovation, it doesn't thoroughly demonstrate its effectiveness through controlled experiments. The comparison with SeqKD might be not enough because SeqKD's lower performance might be attributed to the authors' implementation choice of using only 768 tokens for approximation.
- Furthermore, the omission of comparisons with recent work in small language models, particularly MobileLLM and its associated baselines in the 200M/500M parameter range, leaves an important gap in understanding how MiniPLM compares to state-of-the-art approaches in efficient model training.

[MobileLLM] MobileLLM: Optimizing Sub-billion Parameter Language Models for On-Device Use Cases

**Questions:**

- While the cross-model family experiments with Llama 3.1 and Mamba are interesting, they lack details about model configurations, training data, and experimental setup. Could the authors provide them?

---

> ### Author Response · Authors · 2024-11-22
> **Response to Reviewer 25K6 (part 1/2)**
>
> We sincerely thank the reviewer for acknowledging the novelty of our *Difference Sampling* method, its significant storage efficiency, and the clear computational gains it provides. We also greatly appreciate the reviewer’s recognition of the solid theoretical justification and detailed analysis supporting our approach.
>
> ## The effectiveness of knowledge transfer (weakness #1)
>
> We thank the reviewer for the thoughtful feedback. Our teacher model, the official Qwen-1.8B, is based on a non-public large-scale pre-training corpus, making a direct comparison with our pre-trained models inherently unfair.
>
> To demonstrate the effectiveness of knowledge transfer in a fair setting, we eliminate the impact caused by the differences in pre-training corpora and distill a 500M LM from a **1.2B teacher LM pre-trained on our data** (Due to time constraints during the rebuttal period, we could not pre-train a 1.8B teacher LM on our data. Therefore, we used the previously pre-trained 1.2B model as the teacher LM.). In this controlled setting, the 500M student model achieves performance comparable to the teacher LM. The results, shown in the table below, demonstrate that MiniPLM effectively preserves most of the knowledge from the teacher LM.
>
> |                  | HS   | LAM  | Wino | OBQA | ARC-e | ARC-c | PIQA | SIQA | Story | Avg  |
> | ---------------- | ---- | ---- | ---- | ---- | ----- | ----- | ---- | ---- | ----- | ---- |
> | Teacher (1.2B)   | 39.4 | 44.5 | 51.8 | 28.4 | 46.0  | 25.7  | 67.0 | 39.5 | 62.2  | 44.9 |
> | Pre-train w/o KD | 35.8 | 40.1 | 51.0 | 30.2 | 41.7  | 24.4  | 65.4 | 38.2 | 61.4  | 43.2 |
> | MiniPLM          | 38.0 | 42.0 | 52.1 | 30.4 | 45.8  | 25.0  | 66.4 | 38.8 | 61.7  | 44.5 |
>
> ## Larger model gap for distillation (weakness #2)
>
> Due to limit of computational resources, we are unable to complete the distillation of Llama3-8B to 1B within the rebuttal period, as this requires pre-training 1B models from scratch. As a supplement to the experimental results achievable within our computational budget, we provide results for distilling a 4B Qwen-1.5 model into a 500M model, which similarly represents an 8x model size reduction.
>
> |                  | HS       | LAM      | Wino     | OBQA     | ARC-e    | ARC-c    | PIQA     | SIQA     | Story    | Avg      |
> | ---------------- | -------- | -------- | -------- | -------- | -------- | -------- | -------- | -------- | -------- | -------- |
> | Pre-train w/o KD | 35.8     | 40.1     | 51.0     | 30.2     | 41.7     | 24.4     | 65.4     | 38.2     | 61.4     | 43.2     |
> | Vanilla KD       | 36.8     | 39.9     | 52.0     | 29.2     | 44.3     | 24.1     | 65.3     | 37.8     | 61.3     | 43.4     |
> | MiniPLM          | **38.9** | **42.1** | **52.1** | **31.0** | **44.4** | **24.8** | **66.4** | **39.7** | **62.6** | **44.7** |
>
> ## Controlled experiments for Difference Sampling (weakness #3)
>
> Both SeqKD and our method use pre-training token lengths of 1024, ensuring a controlled and fair comparison. As described in lines 295–297, we use 768 tokens as the prompt and have the teacher generate the remaining 256 tokens, forming a total length of 1024. To further analyze *Difference Sampling*, we compare it with two alternative sampling strategies: sampling based on (1) large model’s preference: $\text{topk}\left[\log p(x)|x\in D\right]$ and (2) the sample difficulty reflected by the reference model: $\text{topk}\left[-\log p_{\text{ref}}(x)|x\in D\right]$.
>
> |                           | Avg. Performance |
> | ------------------------- | ---------------- |
> | Pre-train w/o KD          | 39.9             |
> | SeqKD                     | 39.7             |
> | Sample difficulty         | 40.2             |
> | Teacher preference        | 37.6             |
> | Difference Sampling (Ours) | **41.3**         |
>
> The results show that combining both signals, as implemented in *Difference Sampling*, is essential to achieve a high performance.

---

> ### Author Response · Authors · 2024-11-22
> **Response to Reviewer 25K6 (part 2/2)**
>
> ## The comparison to other small language models (weakness #4)
>
> We did not compare our model to MobileLLM because the data, model architecture, and training computation differ significantly. MobileLLM employs extensive model architecture optimizations, including searching for the optimal model width and depth as well as implementing weight-sharing strategies. Furthermore, it requires $2.1\times10^{21}$ FLOPs to train the 350M model on 10T tokens, which is 10x  more computation than that we can afford.
>
> Since the purpose of our work is to explore improved knowledge distillation techniques for pre-training language models, we keep all other training configurations, like pre-training data, model architecture, and optimization strategies at conventional settings to ensure a fair and straightforward comparison.
>
> In the following, we show the comparison of our 500M model to the MobileLLM’s associated baselines **trained on the Pile corpus with the similar model sizes and comparable training computation** to ours. MiniPLM outperforms models trained with 7.0x and 2.5x more FLOPs (Bloom-560M, OPT-350M), and achieves results comparable to Pythia-410M, trained with 5.5x more FLOPs.
>
> |                   | Non-embedding parameters | Training FLOPs ($\times 10^{20}$) | HS   | LAM  | Wino | OBQA | ARC-e | ARC-c | PIQA | SIQA | Story | avg. |
> | ----------------- | ------------------------ | ----------------------- | ---- | ---- | ---- | ---- | ----- | ----- | ---- | ---- | ----- | ---- |
> | Bloom-560M        | 303M                     | 10.4                    | 37.0 | 34.1 | 50.6 | 29.2 | 41.8  | 23.5  | 65.2 | 37.5 | 61.3  | 42.2 |
> | Pythia-410M       | 358M                     | 8.22                    | 40.1 | 44.4 | 53.7 | 29.6 | 45.9  | 24.5  | 67.2 | 38.9 | 63.3  | 45.3 |
> | Cerebras-GPT-590M | 513M                     | 0.66                    | 32.3 | 36.4 | 49.6 | 28.2 | 41.3  | 23.6  | 62.8 | 36.1 | 58.9  | 41.0 |
> | OPT-350M          | 324M                     | 3.78                    | 36.7 | 43.1 | 52.2 | 28.0 | 40.3  | 23.9  | 64.7 | 39.3 | 63.1  | 43.5 |
> | Pre-Train w/o KD  | 307M                     | 1.50                    | 35.8 | 40.1 | 51.1 | 30.2 | 41.8  | 24.4  | 65.4 | 38.2 | 61.4  | 43.2 |
> | MiniPLM           | 307M                     | 1.50                    | 39.0 | 42.6 | 52.2 | 30.2 | 45.8  | 24.9  | 67.0 | 39.0 | 62.2  | 44.8 |
>
> We also want to point out that there are quite a few existing works showing that the model architecture optimization is complementary to knowledge distillation or data refinement [1,2,3]. We leave further exploration on combining these techniques to future work.
>
> [1] Compact Language Models via Pruning and Knowledge Distillation. 2024. In NeurIPS.
>
> [2] Sheared llama: accelerating language model pre-training via structured pruning. 2024. In ICLR.
>
> [3] Structured Pruning Learns Compact and Accurate Models. 2022. In ACL.
>
> ## The experimental details of cross model family distillation (the question)
>
> We have included the relevant experimental details in Appendix B of the revised paper.

---

### Official Review · Reviewer_Yqhf · 2024-11-04

**Soundness:** 4
**Presentation:** 4
**Contribution:** 3
**Rating:** 8
**Confidence:** 3

**Summary:**

This paper introduces an approach to knowledge distillation (KD) for pretraining LLMs. The proposed method, MINILLM, reformulates the objective of KD with reverse KL divergence as a reward maximization problem, enhancing the student model's focus on the teacher model's primary knowledge modes through difference sampling. Evaluations further show the efficiency and effectiveness of this approach.

**Strengths:**

- This paper provides decent theoretical foundations
- The contribution and experiments designed for this paper is very clear.
- Evaluations demonstrate improved performance and data efficiency gained from the method.
- The contribution of this paper is important to scenarios when data is insufficient.

**Weaknesses:**

Overall, I don't have many criticism toward this paper.
- There is not enough emphasis on how much of this approach would require computational resources to perform distillation. A discussion on resource consumption, such as training time or memory usage, would help assess its practicality for real-world deployment.
- The complexity of the training process may impose challenges in reproducibility and implementation.

**Questions:**

Please refer to weakness.

---

> ### Author Response · Authors · 2024-11-22
> **Response to Reviewer Yqhf**
>
> We sincerely thank the reviewer for recognizing the theoretical foundations, clarity of contributions and experiments, and the improved performance and data efficiency demonstrated by our method. The reviewer’s positive feedback and lack of significant criticism are highly encouraging.
>
> ## The discussion on resource consumption (weakness #1)
>
> We have added a detailed discussion on resource consumption in Appendix C. To summarize, we train a reference model and conduct teacher and reference model inference on the pre-training corpus, which are performed **only once offline** to distill the teacher LM’s knowledge to multiple students. This computation can be further reduced by 95% using a proxy model for *Difference Sampling* as described in Appendix E. The student model's training-time computation remains identical to that of Pre-Train w/o KD.
>
> ## The complexity of the training process (weakness #2)
>
> To ensure reproducibility, we will open-source all our code, model, and data used in our work. We have uploaded our code to the supplementary material. The overall training process is straightforward:
>
> - Model inference is performed prior to pre-training and is well-supported by existing open-source libraries [1,2].
> - The data sampling process is performed on CPUs, requiring only ~60 lines of code.
> - The pre-training framework for both the student and reference LM is nearly identical to that of Pre-Train w/o KD [2,3], with only minor adjustments to the data loading path.
>
> A particularly noteworthy advantage of MiniPLM is that, once $\log \frac{p(x)}{p_{\text{ref}}(x)}$ values are computed for a given corpus, the teacher LM’s knowledge can be distilled into any future LMs **without requiring access to either the teacher or reference model.** This makes the complexity of applying MiniPLM nearly equivalent to that of Pre-Train w/o KD.
>
> [1] DeepSpeed Inference: Enabling Efficient Inference of Transformer Models at Unprecedented Scale. 2022. arxiv pre-print.
>
> [2] Transformers: State-of-the-Art Natural Language Processing. 2020. In ACL.
>
> [3] Megatron-LM: Training Multi-Billion Parameter Language Models Using Model Parallelism. 2019. arxiv pre-print.

---

> > ### Comment · Reviewer_Yqhf · 2024-11-22
> >
> > Thank you for the response. I remain in favor of accepting this paper.

---

### Official Review · Reviewer_ZWq4 · 2024-11-05

**Soundness:** 3
**Presentation:** 3
**Contribution:** 3
**Rating:** 6
**Confidence:** 4

**Summary:**

The paper studies the field of language model pre-training by proposing an efficient KD framework. MINIPLM’s use of offline inference and Difference Sampling stands out as a method that balances computational efficiency and model performance improvement. The findings are well-supported by robust experiments, although the explanations and practical considerations could be more user-friendly and extensive.

**Strengths:**

The introduction of MINIPLM, which uses offline teacher inference and a unique Difference Sampling method, provides an innovative solution to enhance KD during pre-training. This method effectively balances computational costs and improves student language model performance.

The paper presents extensive experimental results across a variety of downstream tasks and model sizes (200M, 500M, 1.2B parameters), demonstrating the effectiveness of MINIPLM. Comparisons with various baselines show substantial improvements.

The results on computation reduction and performance scalability (e.g., 2.2x computation reduction) validate the practicality of MINIPLM for large-scale pre-training.

**Weaknesses:**

The explanation of Difference Sampling and its theoretical justification could be simplified or supplemented with more intuitive examples for clarity.

The requirement for teacher LM’s probabilities on pre-training corpus data may pose challenges for closed-source models or environments with limited computational resources. This could be highlighted more explicitly as a practical limitation.

The paper would benefit from a more detailed discussion on the transferability and generalizability of MINIPLM across different language models and types of downstream tasks, beyond the ones tested.

While a brief limitations section is included, expanding on potential drawbacks, such as cases where MINIPLM may underperform compared to online KD or specific model configurations where it might not be ideal, would strengthen the discussion.

**Questions:**

The explanation of Difference Sampling and its theoretical justification could be simplified or supplemented with more intuitive examples for clarity.

The paper should discuss in detail the limitations of the proposed method.

The transferability and generalizability of MINIPLM across different language models and types of downstream tasks should be discussed.

---

> ### Author Response · Authors · 2024-11-22
> **Response to Reviewer ZWq4**
>
> We sincerely thank the reviewer for acknowledging the strengths of our work, including the innovation of MiniPLM with its *Difference Sampling* method, the extensive experimental validation, and the practical benefits in computation reduction and performance scalability.
>
> ## Explanation of Difference Sampling (weakness #1 and question #1)
>
> We thank the reviewer for the constructive suggestion. In Appendix F, we provide the intuitive examples from our pre-training corpus together with a  detailed discussion to further clarify the *Difference Sampling* mechanism, and add a cross reference to this information in lines 238-240 of the revised manuscript.
>
> ## Distillation from closed-source models (weakness #2 and question #2)
>
> We discussed this limitation in Section 5 and provided a simple solution at the expense of a large number of API calls.  Furthermore, we have explicitly clarified in lines 77–79 of the revised manuscript that our work focuses on knowledge distillation in scenarios where the teacher LM's probabilities are accessible, such as in the open-source or white-box settings.
>
> ## Transferability and generalizability of MiniPLM (weakness #3 and question #3)
>
> MiniPLM is designed to impose minimal constraints on data and models, enabling its application across a wide range of language models trained with cross-entropy loss, just like other works on knowledge distillation [1,2,3]. In addition, in Table 3, we show that MiniPLM generalizes to the setting of knowledge distillation across model families. Since MiniPLM operates during pre-training, the distilled model retains compatibility with downstream tasks, which are typically handled by conventionally pre-trained models without knowledge distillation.
>
> [1] Compact Language Models via Pruning and Knowledge Distillation. 2024. In NeurIPS.
>
> [2] MiniLLM: Knowledge Distillation of Large Language Models. 2024. In ICLR.
>
> [3] On-policy distillation of language models: Learning from self-generated mistakes. 2024. In ICLR.
>
> ## The discussion of MiniPLM’s limitation (weakness #4 and question #2)
>
> One possible limitation of MiniPLM is that *Difference Sampling* samples data points individually. This means that if the original corpus contains too many duplicates, Difference Sampling will also select duplicated samples. Therefore, MiniPLM is better suited for deduplicated data, like many recent pre-training corpora [1,2,3,4].
>
> [1] The Pile: An 800GB Dataset of Diverse Text for Language Modeling.
>
> [2] The RefinedWeb Dataset for Falcon LLM: Outperforming Curated Corpora with Web Data, and Web Data Only.
>
> [3] SemDeDup: Data-efficient learning at web-scale through semantic deduplication. 2023. In ICLR Workshop.
>
> [4] D4: Improving LLM Pretraining via Document De-Duplication and Diversification. 2023. In NeurIPS.

---

### Author Response · Authors · 2024-11-25
**A gentle reminder to the reviewers**

We sincerely thank the reviewers for their thoughtful comments and constructive suggestions, which have greatly helped us improve our paper. We are encouraged to see that the reviewers recognize the novelty (Reviewers ZWq4, 25K6), elegance (Reviewer Rr1c), decent theoretical foundation of our method MiniPLM (Reviewers Yqhf, 25K6), the extensive and solid experiments (Reviewers ZWq4, Yqhf, 25K6, Rr1c, rC3o), and the clarity of our presentation (Reviewers Yqhf, rC3o).

We have provided thorough responses and additional results to address the raised concerns. As we are approaching the end of the discussion stage, we would greatly appreciate it if the reviewers could review our responses and consider updating their scores if the concerns have been addressed. We are happy to engage in further discussion on any remaining issues.

Thank you once again for your time and valuable feedback!

---

### Meta-Review · Area_Chair_TE3n · 2024-12-21

**Metareview:**

The paper introduces MINIPLM, a method for simplifying large language models. MINIPLM uses ``Difference Sampling'' to find difficult training examples where a large teacher model and a smaller reference model give different answers. When trained under the same conditions, MINIPLM performs better than other methods and works well with different model sizes.

Reviewer ZWq4 and Reviewer Yqhf find the method and experimental results clear, highlighting the effectiveness of MINIPLM. Reviewer 25k6 notes the computational efficiency gains as a key advantage, while Reviewer rC3o commends the paper for being easy to follow.
Reviewers 25K6 and Rr1c raised concerns about the practicality. They questioned the efficiency of the method and the criteria for choosing the small reference model. Reviewer 25K6 also questioned the offline inference process and how it affects resource usage. Additionally, Rr1c asked for more proof that the reference model helps improve performance.

During the rebuttal, the authors made great efforts to address these issues. The conducted additional experiments, such as an ablation study on different reference models, teacher model size, and compared MINIPLM with other open-source small language models.

After the rebuttal, all reviewers reached a consensus to accept the paper. The AC agree with the reviewers and recommended accepting the paper. Nonetheless, AC suggests that the authors include additional experiments and discussions from the rebuttal in the final version of the paper.

**Additional Comments On Reviewer Discussion:**

During the review process, reviewer 25K6 and Rr1c raised concerns about whether the method is practical and effective. They specifically questioned how efficient it is and why a small reference model was chosen. Reviewer 25K6 also wanted more information about the offline inference process and how it affects resource usage. Additionally, Rr1c asked for more proof that the reference model helps improve performance.

The authors made great efforts to address the reviewers’ concerns. They performed abundant additional experiments, such as an ablation study on different reference models, and larger teacher models, and compared MINIPLM to other open-source small language models. After the rebuttal discussion, most concerns were resolved.

Reviewers Yqhf, Rr1c, rC3o confirmed their concerns have been resolved, and all reviewers reached a consensus to accept this paper. The AC agreed with the reviewers and recommended accepting this paper.

---

### Decision · Program_Chairs · 2025-01-22

Accept (Poster)